# Fast Tree-Field Integrators: From Low Displacement Rank to Topological Transformers

**Krzysztof Choromanski**[1,2][*], **Arijit Sehanobish**[3,*], **Somnath Basu Roy Chowdhury**[4,*],
**Han Lin**[4,*], **Avinava Dubey**[5,*], **Tamas Sarlos**[5], **Snigdha Chaturvedi**[4]

[1] Google DeepMind, [2] Columbia University, [3] Independent, [4] UNC Chapel Hill, [5] Google Research.

## Abstract

We present a new class of fast polylog-linear algorithms based on the theory of structured matrices (in particular *low displacement rank*) for integrating tensor fields defined on weighted trees. Several applications of the resulting *fast tree-field integrators* (FTFIs) are presented, including (a) approximation of graph metrics with tree metrics, (b) graph classification, (c) modeling on meshes, and finally (d) *Topological Transformers* (TTs) [Choromanski et al., 2022] for images. For Topological Transformers, we propose new relative position encoding (RPE) masking mechanisms with as few as **three** extra learnable parameters per Transformer layer, leading to **1.0-1.5%+** accuracy gains. Importantly, most of FTFIs are **exact** methods, thus numerically equivalent to their brute-force counterparts. When applied to graphs with thousands of nodes, those exact algorithms provide **5.7-13x** speedups. We also provide an extensive theoretical analysis of our methods.

## 1 Introduction

Matrix-vector multiplication remains a key computational block of virtually all modern machine learning (ML) algorithms. For this reason, decades of research have been dedicated towards making this fundamental operation more efficient. One approach to achieve this goal is through efficient hardware design, e.g., using modern GPU and TPU accelerators [Abadi et al., 2016, Yu et al., 2022, 2020]. The alternative method involves developing algorithms for efficient matrix-vector multiplication by leveraging either (1) sparse matrices [Wang, 2021, Beniamini et al., 2020], or (2) structured dense matrices [Thomas et al., 2018, Chandrasekaran et al., 2018]. These algorithms can be applied in modern neural network systems, where weights are pruned to encourage sparsity [Blalock et al., 2020] or they can be parameterized with structured matrices [Sindhwani et al., 2015].

In this work, we aim to accelerate multiplications with a large class of matrices, that we refer to as *f-distance matrices*, which play an important role in several ML algorithms. Consider a matrix $\mathbf{M}_f^{\mathrm{G}} = [f(\mathrm{dist}(i,j))]_{i,j=1,...,N} \in \mathbb{R}^{N \times N}$, where $\mathrm{dist}(i,j)$ stands for the shortest-path distance between the $i$-th and $j$-th vertex of an undirected graph $\mathrm{G} = (\mathrm{V}, \mathrm{E}, \mathrm{W})$. Here $\mathrm{V} = \{1, ..., N\}$ stands for the set of vertices (nodes), E denotes the set of edges, $\mathrm{W} : \mathrm{E} \to \mathbb{R}_+$ maps them to their positive weights, and $f : \mathbb{R} \to \mathbb{R}$. We call $\mathbf{M}_f^{\mathrm{G}}$ a *f-distance matrix in* G. Note that if $f(x) \stackrel{\text{def}}{=} x$, then $\mathbf{M}_f^{\mathrm{G}}$ is the Shortest Path Kernel matrix.

The product $\mathbf{M}_f^{\mathrm{G}}\mathbf{x}$ (where $\mathbf{x} \in \mathbb{R}^N$) represents a scalar field on V obtained by discretely integrating the field defined by $\mathbf{x}$. In this integration, a new field value at a vertex $v$ is calculated by averaging the old field values at all vertices $u$, weighted according to the function $f(\mathrm{dist}(v,u))$. This integration can

---

[*]equal contribution

38th Conference on Neural Information Processing Systems (NeurIPS 2024).

be extended to general tensor fields by replacing vector $\mathbf{x} \in \mathbb{R}^N$ with a tensor $\mathbf{X} \in \mathbb{R}^{N \times d_1 \times d_2 \times \cdots}$:

$$\mathbf{M}_f^{\mathrm{G}} \mathbf{X}[i] = \sum_{j \in \mathrm{V(G)}} f(\mathrm{dist}(i,j)) \mathbf{X}[j] \tag{1}$$

We refer to the above procedure as the $f$-integration of a field $\mathbf{X}$ on G. We will use the terms *graph field integration* (GFI) and *multiplication with $f$-distance matrices* interchangeably throughout the paper. When the graph, G, is a tree, we call this procedure (Eq. 1) *tree field integration*. Next, we highlight several applications that rely on multiplications with $f$-distance matrices, $\mathbf{M}_f^{\mathrm{G}}$.

1. **Interpolation on manifolds:** This task involves predicting unseen values on a manifold from a set of known values. For example, predicting the velocities of all points on a flag with known velocities for a few points [Pfaff et al., 2021]. For a discretized manifold, the interpolated values can be obtained using a weighted average using graph field integration (Eq. 1).

2. **Optimal Transport (OT):** A popular method used to solve the entropic OT problem [Peyré and Cuturi, 2019] is the Sinkhorn algorithm [Eckstein and Nutz, 2022]. Sinkhorn relies on multiplications with *cost matrices*, which are special cases of $f$-distance matrices for metric spaces induced by shortest-path distances in graphs. This can be efficiently solved using graph field integration.

3. **Topological Transformers (TTs):** Topological Transformers [Choromanski et al., 2022] are extensions of traditional Transformers [Vaswani et al., 2017] for graph inputs. TTs modify the 1-D relative positional encoding (RPE) using "mask matrices", which are $f$-distance matrices. We show how these matrices can be efficiently integrated into the attention mechanism (Sec. 4.4).

In the above applications, apart from the graph field integration step, the bottleneck lies in the process of explicitly materializing the $f$-distance matrix. Naively performing the integration in Eq 1 consists of two steps: **(a)** computing the $f$-distance matrix, $\mathbf{M}_f^{\mathrm{G}}$, which requires $O(N^3)$ time in the worst case (which we call *preprocessing*), and **(b)** performing the multiplication takes $O(N^2)$ time. This is prohibitively expensive while using large graphs.

In this paper, we introduce a new class of fast polylog-linear algorithms for graph field integration that uses low displacement rank (LDR) matrices [Thomas et al., 2018, Chandrasekaran et al., 2018]. To summarize, our primary contributions are given below:

1. We provide the first **exact** polylog-linear multiplication algorithms called **Fast Tree-Field Integrators** (FTFIs), for general weighted trees and a rich class of maps $f$, including rational, trigonometric, exponential and exponentiated quadratic functions (Sec. 3.2).

2. We show how Fast Tree-Field Integrators can be applied to support fast computations on general graphs by approximating graph metrics with tree metrics (Sec. 4).

3. We show that FTFIs are **5.7-10x** faster than baseline graph field integration methods for large-scale graphs (Sec. 4.1 and 4.2).

4. We showcase the efficacy of FTFIs in several applications including graph classification (Sec. 4.2), interpolation on meshes (Sec. 4.2), and Topological Vision Transformers (TVTs) (Sec. 4.4). For TVTs, we propose new relative position encoding (RPE) masking mechanisms by introducing only **three** extra learnable parameters, which leads to **1.0-1.5%** accuracy gains. We provide an exhaustive evaluation on Vision Performers (**25** models on multiple datasets). Some of our best models use exponentiated quadratic functions $f$, which has not been applied in this context before.

For completeness, we also propose approximate FTFI extensions via *Non-Uniform FFT* (NU-FFT) [Kircheis et al., 2023] and random Fourier features (RFFs) [Rahimi and Recht, 2007] (Sec. A.2).

## 2   Related work

Efficient graph field integration (Eq. 1) has been studied by prior works for different classes of matrices. For example, Al-Mohy and Higham [2011] considered exponentiated adjacency matrix-vector multiplication, Spielman and Teng [2012] targeted symmetric diagonally dominant matrices (e.g., Laplacian), Arrigo et al. [2018] analyzed matrices that are power series of random walk kernels. In contrast to these approaches, Saad and Schultz [1986] proposed general iterative methods for solving certain linear systems using Arnoldi's iterations. However, These iterative methods can suffer

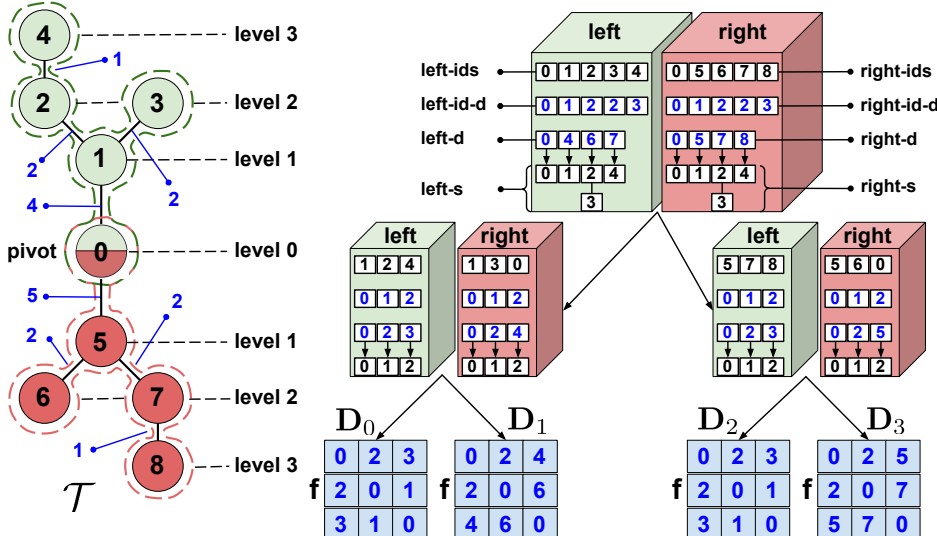

Figure 1: Pictorial representation of the IntegratorTree (see: Sec 3.1) data structure for the nine-vertex input tree $\mathcal{T}$ on the left. Numbers in blue next to the input tree denote the weights of its edges. Leaves of the IntegratorTree object represent $f$-transformed (element-wise) distance matrices: $\mathbf{D}_0, \mathbf{D}_1, \mathbf{D}_2, \mathbf{D}_3$ for sub-trees induced by vertex-sets: $\{1, 2, 4\}, \{1, 3, 0\}, \{5, 7, 8\}$ and $\{5,6,0\}$ respectively. Different *levels* correspond to different distances from the pivot point.

from convergence issues. Williams [2007] showed that it is possible to pre-process any boolean matrix to achieve sub-quadratic matrix-vector multiplication.

The general problem of computing the action of a matrix on a vector, where the matrix is the graph kernel, in sub-quadratic time is intractable, except for a few special cases [Al-Mohy and Higham, 2011, Choromanski et al., 2023]. In this work, we embed the graph G under consideration in a tree (replacing the graph metric by the underlying *tree metric*). Then, we leverage the tree structure to approximate the action of the kernel on a given vector by providing **exact** integration on a tree.

Previous works [Bartal et al., 2022, 2019, Abraham et al., 2008, Bartal, 1998] have used the theory of *tree metrics* (TMs) in several applications in mathematics and computer science. TMs are widely used to embed a complex metric space (e.g., a Riemannian manifold) into a more tractable one, while approximately preserving (all or most of the) pairwise distances. They find applications in distributed & online algorithms [Khan et al., 2008, Bubeck et al., 2018], biology [Mossel, 2007], vision, robotics [Athitsos and Sclaroff, 2003], and ML (e.g., metric spaces' regression [Gottlieb et al., 2011]).

**Tree metrics for fast matrix multiplication:** Applying tree metrics (TM) to compute approximate $\mathbf{M}_f^{\mathrm{G}}$ is a natural approach to scale up matrix multiplications. If a TM approximates the metric space well, then the derived embeddings should have low distortion. However, in the worst-case scenario, this is not true for deterministic *tree embeddings*. A natural alternative is to sample trees from probabilistic distributions, which are shown to provide logarithmic distortion in expectation [Fakcharoenphol et al., 2004b, Bartal et al., 2022]. This can be further improved to constant distortion for certain classes of metrics, e.g., celebrated *snowflake metics* [Leeb, 2016]. For graph metrics defined by shortest-path distances, there exist spanning trees providing constant average distortion (over all pairs of nodes). These spanning trees can be constructed as *near minimum weight spanning trees* [Bartal et al., 2016]. Unfortunately, explicit application of *any* tree metric still requires $O(N^2)$ time (impractical for large $N$) to: **(1)** compute all shortest-path distances via the breadth-first-search algorithm (BFS), even if sub-quadratic methods were used to construct a tree (e.g. minimum spanning tree), **(2)** store the matrix, and **(3)** perform matrix-vector multiplications. We provide more details about work related to graph field integration in Appendix B.

## 3 Fast Tree-Field Integrators (FTFI)

In this section, we present our approach for performing efficient field integration on a tree, which we call *fast tree field integrator*. We begin by introducing the concept of integrator trees (ITs), which is a

specialized decomposition of a tree using the theory of *balanced separators* (Sec 3.1). Subsequently, we leverage these integrator trees to execute efficient integration on a tree via a *divide-and-conquer algorithm* (Sec 3.2).

## 3.1 IntegratorTrees (ITs) - preliminaries

To support fast integration for various tensor fields $\mathbf{X} \in \mathbb{R}^{N \times d_1 \times \ldots \times d_s}$ defined on a given input tree $\mathcal{T}$, we first design a special data structure that we refer to as an *IntegratorTree* (IT). An object of this type is constructed only once per $\mathcal{T}$, regardless of the number of tensor fields used. An IT is a rooted binary tree. To avoid confusion, we will refer to its vertices as *nodes*, reserving term *vertices* for those of $\mathcal{T}$. Each node of IT corresponds to the induced sub-tree $\mathcal{ST}$ of $\mathcal{T}$. For every non-leaf node corresponding to some $\mathcal{ST}$, a *pivot* point $p$ along with two sub-trees: $\mathcal{ST}_{\text{left}}$ and $\mathcal{ST}_{\text{right}}$ are constructed. The following needs to be satisfied:

- $|\mathcal{ST}_x| \geq \frac{|\mathcal{ST}|}{4}$ for $x \in \{\text{left}, \text{right}\}$,
- $\mathcal{ST}_x \cap \mathcal{ST}_y = \{p\}$ ($|\cdot|$ denotes the number of vertices).

The next lemma shows that every tree $\mathcal{K}$ with $|\mathcal{K}| \geq 6$ has the above decomposition and it can be efficiently found.

**Lemma 3.1** (**Pivoting**). *If $\mathcal{K}$ is a tree with $|\mathcal{K}| \geq 6$, then $\mathcal{K}$ admits a decomposition $(\mathcal{K}_{\text{left}}, \mathcal{K}_{\text{right}}, p)$ given above and it can be constructed in **linear** time.*

The algorithmic proof is provided in Appendix A.1 and uses standard tools from the theory of balanced separators.

The *left child* of the non-leaf node for $\mathcal{ST}$ corresponds to $\mathcal{ST}_{\text{left}}$ and the *right child* to $\mathcal{ST}_{\text{right}}$. In addition to these two pointers, a non-leaf node also contains eight extra fields, partitioned into two groups, one corresponding to its left child and one to its right children. The fields corresponding to the left child are as follows:

- **Left-ids:** an array of the ids (in $\mathcal{T}$) of those vertices that are in $\mathcal{ST}_{\text{left}}$, mapping the ids of vertices in $\mathcal{ST}_{\text{left}}$ to the original ids in $\mathcal{T}$ (each sub-tree uses consecutive numbers from 0 as ids locally).
- **Left-d:** an array of different shortest-path **d**istances from the pivot point to the vertices in $\mathcal{ST}_{\text{left}}$.
- **Left-id-d:** an array mapping the ids of vertices (in $\mathcal{ST}_{\text{left}}$) to the indices in left-d of their corresponding distances from the pivot point.
- **Left-s:** a corresponding array of the ordered sub-**s**ets of ids (in $\mathcal{ST}_{\text{left}}$) of vertices within a particular distance from the pivot point.

Fields corresponding to the right child are defined similarly. The leaf nodes of the IT consist only of the $f$-transformed (element-wise) distance matrices $\mathbf{D}$ for their corresponding sub-trees (see: Fig 1). In principle, the leaf nodes of IT correspond to sub-trees with less than $t = 6$ vertices each. In practice, we choose higher $t$, for more efficient integration (see: discussion in Sec. 4.1).

**Time & space complexity of constructing ITs:** From what we have said so far, it is clear that an IT can be constructed by applying *breadth first search* (BFS) and the linear algorithmic procedure for constructing the decomposition from Lemma 3.1. Note that every vertex of the input tree appears in the logarithmic number of nodes in the IT since the size of the sub-tree is at most $\frac{3}{4} \times$ the size of its parent in IT. We conclude that IT for the given input tree $\mathcal{T}$ can be computed in $O(N \log(N))$ time, where $N$ stands for the number of vertices $|\mathcal{T}|$ of $\mathcal{T}$.

## 3.2 Integrating with IntegratorTrees

We are ready to explain how ITs allow us to efficiently integrate any given tensor field $\mathbf{X} \in \mathbb{R}^{N \times d_1 \times \ldots \times d_s}$ defined on $\mathcal{T}$ for a wide class of function $f : \mathbb{R} \to \mathbb{R}$. We will apply a *divide-and-conquer* strategy.

We start in the root node of IT. If that node is a leaf then the $f$-transformed distance matrix is stored and can be directly used for matrix-tensor multiplication. If this node is not a leaf, then it encodes the decomposition $(\mathcal{T}_{\text{left}}, \mathcal{T}_{\text{right}}, p)$. Take some $v \in \mathrm{V}(\mathcal{T}_{\text{left}})$. Note that the value $\mathbf{M}_f^G \mathbf{X}[v]$ of the new

field in $v$ after $f$-integration is given as follows for $\mathcal{W} = \mathrm{V}(\mathcal{T}_{\text{right}}) \setminus \{p\}$:

$$\underbrace{\sum_{j \in \mathrm{V}(\mathcal{T}_{\text{left}})} f(\text{dist}(v,j)) \mathbf{X}[j]}_{\mathrm{F}_{\text{inner}}(v)} + \underbrace{\sum_{j \in \mathcal{W}} f(\text{dist}(v,j)) \mathbf{X}[j]}_{\mathrm{F}_{\text{cross}}(v)}. \tag{2}$$

To compute the new values of the field for nodes $v \in \mathcal{T}_{\text{left}}$, one needs to:

1. Compute the contribution to it from $\mathcal{T}_{\text{left}}$ ($\mathrm{F}_{\text{inner}}(v)$-terms). This can be done simply by applying Eq. 2 recursively for $\mathcal{T}_{\text{left}}$, which means traversing to the left child of the root.

2. Add the so-called *cross-terms* contributions coming from the vertices of $\mathcal{W}$ ($\mathrm{F}_{\text{cross}}(v)$-terms).

The key observation is that the latter (cross-term) contributions can be retrieved simply by computing $\mathbf{CX}'$, where: (1) $\mathbf{C} \in \mathbb{R}^{k \times l}$ with $k$ and $l$ being the sizes of the node's left-d and right-d arrays respectively. $\mathbf{C}(i,j) = f(\text{left-d}[i] + \text{right-d}[j])$, and (2) Let $b_j \overset{\text{def}}{=} |\text{right-s}[j]|$ where $|\cdot|$ refers to the size of the subset. Then $\mathbf{X}' \in \mathbb{R}^{l \times d_1 \times \dots \times d_s}$ is defined as follows:

$$\mathbf{X}'[j] \overset{\text{def}}{=} \sum_{z=0}^{b_j - 1} \mathbf{X}[\text{right-ids}[\text{right-s}[j][z]]]. \tag{3}$$

Given the structure of IT, tensor $\mathbf{X}'$ can be computed in linear time. Note that the following holds:

$$\mathrm{F}_{\text{cross}}(v) = (\mathbf{CX}')[\tau(v)] - f(\text{left-d}[\tau(v)])\mathbf{X}'[0], \tag{4}$$

where $\tau(v) = \text{left-id-d}[v]$. Analogous analysis can be derived for $v \in \mathcal{T}_{\text{right}}$, with matrix $\mathbf{C}^\top$ replacing $\mathbf{C}$. Thus the overall time complexity of the cross-terms computations is determined by the algorithm for matrix-tensor multiplications with matrices $\mathbf{C}$ and $\mathbf{C}^\top$.

### 3.2.1 The case for structured matrices: multiplications with $\mathbf{C}, \mathbf{C}^\top$ and cordiality

Matrices $\mathbf{C}, \mathbf{C}^\top$ are of the form: $[f(x_i + y_j)]_{i=1,\dots,a}^{j=1,\dots,b}$ for some sequences $X = (x_i)_{i=1}^a$, $Y = (y_j)_{j=1}^b$ and $a, b \in \mathbb{N}_+$.

**Definition 3.2** (cordial functions). A function $f : \mathbb{R} \to \mathbb{R}$ is *d-cordial* (or: *cordial* if $d$ is not specified), if there exists $d \in \mathbb{N}$ such that matrix-vector multiplication with a matrix $\mathbf{M} = [f(x_i + y_j)]_{i=1,\dots,a}^{j=1,\dots,b}$ can be conducted in time $O((a+b)\log^d(a+b))$ for every $(x_i)_{i=1}^a, (y_j)_{j=1}^b$.

Next, we demonstrate the importance of cordial functions in our FTFI framework.

**Lemma 3.3** ($f$-integration with cordial functions). *If $f$ is $d$-cordial then $f$-integration for the general weighted tree of $N$ vertices can be conducted in time $O(N \log^{d+1}(N))$.*

*Proof.* Denote by $T(N)$ time complexity for running FTFI on the $N$-vertex tree. We have the following recursive formula for $T$, where $\frac{1}{4} \le c \le \frac{3}{4}$:

$$T(N) \le T(cN) + T((1-c)N) + O(N \log^d(N)) \tag{5}$$

This is implied by the fact that: (1) the size of each sub-tree is at most $\frac{3}{4} \times$ the size of its parent, (2) the computation across left and right children is dominated by multiplications with matrices $\mathbf{C}$ and $\mathbf{C}^\top$. The solution of this recursion leads to the statement. $\square$

Next, we show some practical implications of Lemma 3.3, where tree weights are **completely arbitrary**. Additional results are given in Sec. A.2.3.

**Rational functions:** We claim that every rational $f$ is $(2 + \epsilon)$-cordial for any $\epsilon > 0$. We will use Lemma 1 from [Cabello, 2022] stating that: given any set of $b$ rational functions $R_j(x) = \frac{P_j(x)}{Q_j(x)}$ and $\{x_i\}_{i=1}^a$, one can compute the $a$ values $\sum_{j=1}^b R_j(x_i)$ in time $O((a+b)\log^2(b)\log(\log(b)))$ (by applying FFT). For a given vector $\mathbf{v} \in \mathbb{R}^b$, it thus suffices to define: $R_j(x) = v_j f(x + y_j)$ and that lemma can be applied to efficiently compute $\mathbf{Mv}$. We conclude that for any $\epsilon > 0$, $f$-integration can be conducted in $O(N \log^{3+\epsilon}(N))$ time for $N$-vertex weighted trees and any rational $f : \mathbb{R} \to \mathbb{R}$ (see also: Sec. 4.3, Sec. 4.2, Sec. 4.4).

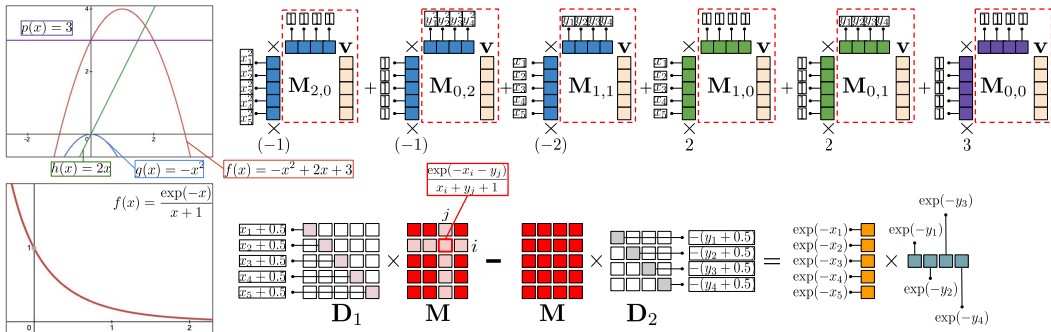

Figure 2: Pictorial representations of the main concepts behind efficient matrix-vector multiplications $\mathbf{Mv}$ with $\mathbf{M} \in \mathbb{R}^{5 \times 4}$, for the polynomial $f$ and $f(x) = \frac{\exp(\lambda x)}{x+c}$. In the polynomial case, $\mathbf{M}$ is re-written as a sum of low-rank outer-product matrices corresponding to terms of different degrees (e.g., constant, linear, quadratic, etc.). Matrix associativity property is applied for efficient calculations (dotted-border blocks indicating the order of computations). In the second case, $\mathbf{M}$ is high-rank, but the so-called *low displacement rank operator* $\Delta_{D_1, D_2} : \mathbf{X} \rightarrow \mathbf{D_1 M} - \mathbf{M D_2}$ for diagonal $\mathbf{D_1}, \mathbf{D_2}$ can be applied to make it a low-rank outer-product matrix. The multiplication with $\mathbf{M}$ can be efficiently performed using the theory of LDR matrices [Thomas et al., 2018].

**Polynomial functions:**    The above result on rational functions clearly applies also to polynomial $f$, but here we can do better. We show that $f$ is 0-cordial. Assume that $f(x) = \sum_{t=0}^{B} a_t x^t$. We have: $\mathbf{M} = \sum_{t=0}^{B} \sum_{l=0}^{t} a_t \binom{t}{l} \mathbf{M}_{l, t-l}$, where matrix $\mathbf{M}_{u,v} \in \mathbb{R}^{a \times b}$ is defined as an outer-product of two vectors: $(x_1^u, ..., x_a^u) \in \mathbb{R}^a$ and $(y_1^v, ..., y_b^v) \in \mathbb{R}^b$. Thus each $\mathbf{M}_{u,v}$ supports linear matrix-vector multiplication (via associativity property). The proof is completed, since $B$ is a constant. We conclude that $f$-integration can be conducted in $O(N \log(N))$ time for $N$-vertex weighted trees and any polynomial $f : \mathbb{R} \rightarrow \mathbb{R}$ (see: Fig. 2 and Fig 9).

**Exponential functions:**    Take $f(x) = \exp(\lambda x)$. Then $\mathbf{M}$ is an outer-product of two vectors: $(\exp(\lambda x_i))_{i=1}^{a} \in \mathbb{R}^a$ and $(\exp(\lambda y_j))_{j=1}^{b} \in \mathbb{R}^b$. The remaining analysis and conclusion is thus the same as for the polynomial case (see also: Sec. 4.4).

**Function:** $f(x) = \frac{\exp(\lambda x)}{x+c}$**:**    ($c$ is a constant) We claim that $f$ is 2-cordial. In that setting, matrix $\mathbf{M}$ satisfies: $\mathbf{M}(i,j) = \frac{\exp(\lambda x_i) \exp(\lambda y_j)}{(x_i + \frac{c}{2}) + (y_j + \frac{c}{2})}$ and thus is a *Cauchy-like* LDR, supporting fast $O(N \log^2(N))$ matrix-vector multiplication [Victor Y. Pan, 2000]. We conclude that $f$-integration can be conducted in $O(N \log^3(N))$ time for $N$-vertex weighted trees and $f(x) = \frac{\exp(\lambda x)}{x+c}$ (see: Fig. 2).

**Functions $f(x) = \exp(ux^2 + vx + w)$ and trees with positive rational weights:**    Now matrix $\mathbf{M}$ can be re-written as $\mathbf{M} = \exp(w) \mathbf{D_1 V D_2}$, where $\mathbf{D_1} \in \mathbb{R}^{a \times a}$ and $\mathbf{D_2} \in \mathbb{R}^{b \times b}$ are diagonal, with diagonal entries given by sequences $\{\exp(ux_i^2 + vx_i)\}_{i=1}^{a}$ and $\{\exp(uy_j^2 + vy_j)\}_{j=1}^{b}$ respectively, and furthermore $\mathbf{V}$ is the *generalized Vandermonde matrix* (GVM) (using arbitrary nonnegative integers as exponents). It is defined as: $\mathbf{V}(i,j) = r_i^{s_j}$, where $r_i = \exp(\frac{2ux_i}{q})$ and $s_j = y_j q \in \mathbb{N}$. As in the previous case, the embedding trick can be applied, but we will use it only for columns. That effectively leads to the completion of the set of exponents $\{s_j\}$ to the set of consecutive integers starting from 0 and a regular Vandermonde matrix, that supports $O(N \log^2(N))$ matrix-vector multiplication, replacing GVM. The benefit of this embedding, as compared to the previous one, is that even though it still increases the number of columns by a multiplicative factor of $p$, the number of rows does not change. Therefore, for $p \gg \log(N)$, substantial computational speedups are achieved (see: Sec. 4.4).

## 4   Experiments

In this section, we outline the experimental setup and report the performance of FTFI across various settings. For all the experiments, we only consider minimum spanning tree (MST) as an approximation of our graph. Specifically, we design experiments to answer these research questions:

(**Q1**) How efficient are FTFIs for tree field integration?
(**Q2**) How does the approximation quality of FTFI compare to other integration algorithms?
(**Q3**) How can we further improve the approximation quality in FTFI?
(**Q4**) How can we use FTFI in real-world large-scale settings?

## 4.1 Runtime Efficiency of FTFI

The main goal of this experiment is to evaluate the speedups obtained by FTFI as compared to brute-force tree field integrator (BTFI) i.e. the explicit calculation of Eq 1 on a tree. We consider two classes of graphs: **(a)** *synthetic*, obtained from a path-graph by adding random edges and **(b)** *mesh graphs* from Thingi10K [Zhou and Jacobson, 2016] dataset. For BTFI, we compute the MST and then integrate a random scalar field **X** on the vertices of the MST. Since BTFI & FTFI are numerically equivalent, we report the pre-processing time and integration as a function of vertex count ($N$) in Fig. 3. We observe that FTFI achieves up to **13x** speedups for 20K-vertex meshes and **5.7x+** for synthetic graphs with over 10K vertices compared to BTFI.

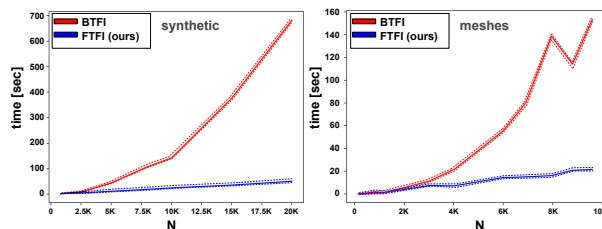

Figure 3: Runtime comparison of FTFI with BTFI as a function of the number of vertices, $N$. **Left:** Synthetic graphs. **Right**: Mesh-graphs from Thingi10K. The speed is not necessarily monotonic in $N$ as it depends on the distribution of lengths of the shortest paths. For each graph, 10 experiments were run (std. shown via dotted lines).

## 4.2 Approximation Quality of FTFI

We evaluate the approximation quality achieved by FTFI across a wide range of graph-based tasks.

**Interpolation on meshes.** We compare the efficiency of FTFI with baselines on the *normal vector prediction task*. Every node of the considered mesh G with a vertex-set V, is associated with a location $\mathbf{x}_i \in \mathbb{R}^3$ and a vertex normal $\mathbf{F}_i \in \mathbb{R}^3$. For each mesh, we randomly select a subset $V' \subseteq V$ with $|V'| = 0.8|V|$ and mask out their vertex normals (set as zero vectors). The interpolation task involves predicting the vertex normals of each masked node $i \in V'$ as: $\mathbf{F}_i = \sum_{j \in V \setminus V'} K_f(i,j)\mathbf{F}_j$, where $K_f(w,v) = f(\text{dist}(w,v))$, with $\text{dist}(w,v)$ being the shortest path distance between node $w$ and $v$, and $f$ is a rational function $f(x) = 1/(1 + \lambda x^2)$. We perform a grid search to set hyperparameter $\lambda$ for each mesh and report the result with the highest cosine similarity between predicted and ground truth vertex normals, averaged over all the nodes. We run tests on **40 meshes** of the 3D-printed objects with a wide range of sizes from the Thingi10K dataset (details in Appendix D.3). We compare FTFI with BTFI, low-distortion tree-based algorithms such as Bartal Trees [Bartal, 1996] and FRT trees [Fakcharoenphol et al., 2004a] alongside the state-of-the-art method for graph-field integration, the Separator Factorization (SF) algorithm [Choromanski et al., 2023]. We also compare against the baseline BGFI which entails explicitly materializing the kernel matrix of G and then performing matrix tensor multiplication with a tensor field $\mathbf{F}$ defined by the $\mathbf{F}_i$'s.

Preprocessing involves building specific tree structures (FRT, Bartal), calculating the kernel matrices (BGFI, BTFI), or creating specialized data structures (SF, FTFI) for efficient later use. The first two plots in Fig. 4 shows the pre-processing time and cosine similarity for various algorithms applied to meshes of different sizes. FTFI is the fastest in terms of pre-processing time and achieves competitive performance in terms of cosine similarity (between predicted and actual vertex normals) when compared with the SF algorithm while being numerically equivalent to BTFI. FTFI is a few orders of magnitude faster than BTFI and the tree-based methods while maintaining accuracy.

**Graph classification.** Graph kernels have been widely used for graph classification tasks in previous works [Kriege et al., 2020, Nikolentzos et al., 2021]. We compare the classification results obtained using the approximate kernel from FTFI with those from the exact SP kernel. In this setting, we use the Shortest Path (SP) kernel, $f(\text{dist}(i,j))$. We perform experiments on a wide range of bioinformatics and social networks datasets like D&D, MUTAG, REDDIT, IMDB, among others. We follow [de Lara and Pineau, 2018] and construct the graph feature for both kernels by using the smallest $k$ eigenvalues ($k$ is a hyperparameter). This feature set is then used for classification, using

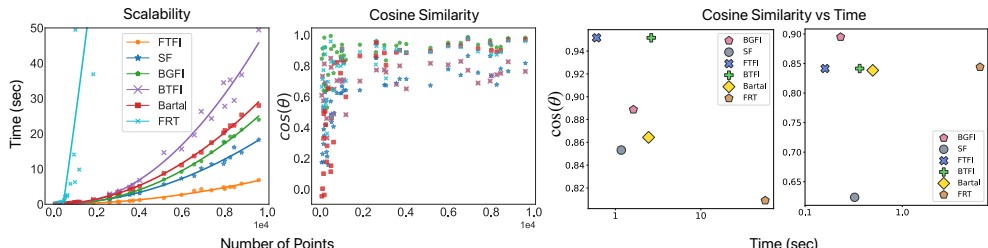

Figure 4: Speed (pre-processing time) and accuracy (cosine similarity) comparison of the FTFI and other baselines for vertex normal prediction on meshes. Cosine similarity of BFFI and FTFI almost overlaps. The last two figures are qualitative examples showcasing the tradeoff between cosine similarity and preprocessing time for meshes of sizes 3K and 5K nodes respectively.

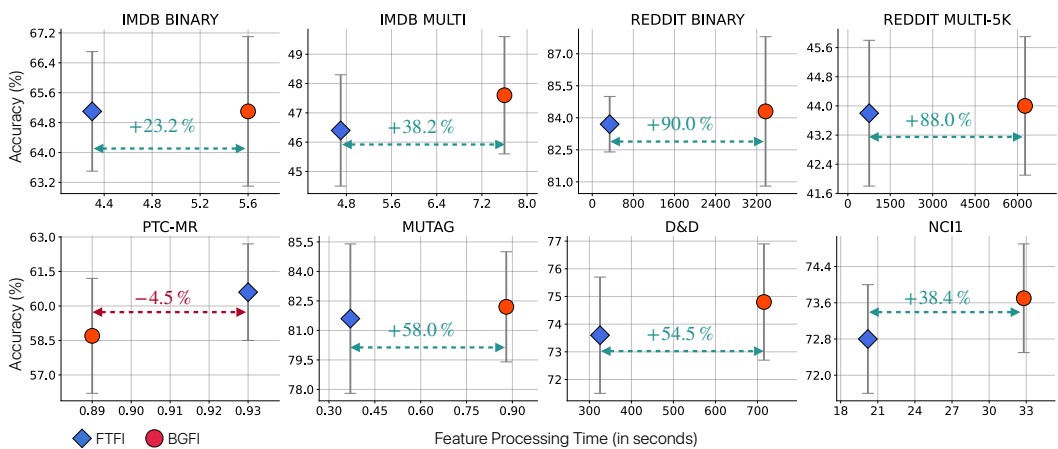

Figure 5: Trade-off plot comparing graph classification accuracy and feature processing time for the classifiers using FTFI and BGFI. FTFI achieves similar accuracy as BGFI while significantly reducing fp time across most datasets. We report the reduction in FTFI's processing time ($\pm$x%) compared to BGFI using a dotted line.

a random forest classifier. We observe that FTFI achieves significant speed improvements while achieving similar accuracy compared to its brute-force counterpart, BGFI (see Fig. 5). We provide more details about the experimental setup and baselines Appendix D.4. We also report additional experiments on meshes and point clouds in Appendix D.1.

### 4.3 Improving approximation quality with learnable $f$-distance matrices

We propose to further improve the approximation quality of FTFI by learning a $f$-distance matrix on metrics derived from the MST. As an application, we choose *general graph metrics*, where our goal is to learn the shortest-path distance $d_{v,w}$ between a given pair of nodes $(v, w)$ in a graph. Given a $f$-distance matrix and tree-derived metric $\widehat{d}_{v,w}$ the objective is to learn a mapping to minimize

$$\mathbb{E}_{(v,w)\in\mathcal{D}}\left[\left(d_{v,w} - f_{b_0,...,b_s}^{a_0,...,a_t}(\widehat{d}_{v,w})\right)^2\right]. \tag{6}$$

Rather than using a fixed $f$, we parameterize and train it. We consider rational function $f$:

$$f_{b_0,...,b_s}^{a_0,...,a_t}(x) = \frac{a_0 + a_1 x + ... + a_t x^t}{b_0 + b_1 x + ... + b_s x^s}, \tag{7}$$

where $a_0, ..., a_t, b_0, ..., b_s \in \mathbb{R}$ are trainable parameters.

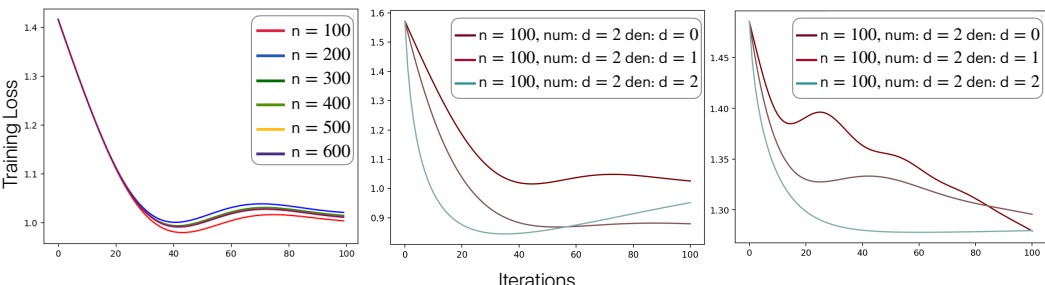

Figure 6: **Left:** Relative Frobenius norm error as a function of the number of training iterations for different sizes $n$ and learnable quadratic $f$. **Middle:** Comparison of the training of different rational functions $f$ with num:d defining the degree of the numerator and den:d, the degree of the denominator for the synthetic graph obtained from a path on $N = 800$ by adding 600 random edges and assigning random weights taken from $(0, 1)$. **Right:** constructed similarly, but for a sampled mesh graphs from Thingi10k dataset.

Training dataset $\mathcal{D}$. For a graph G, we randomly sample vertices. The training dataset consists of tuples of the form: $(v, w, d_{v,w}, \widehat{d}_{v,w}) \in \mathcal{D}$, where $v, w$ are randomly sampled vertices. Each data point can be constructed in time $O(N \log(N))$, or even $O(N)$ if weights are in $\mathbb{N}$ [Thorup, 1997].

Final evaluation. To evaluate the quality of the approximation, we compute the relative Frobenius norm error: $\epsilon = \frac{\|\mathbf{M}_f^{\mathrm{T}} - \mathbf{M}_{\mathrm{id}}^{\mathrm{G}}\|_{\mathrm{F}}}{\|\mathbf{M}_{\mathrm{id}}^{\mathrm{G}}\|_{\mathrm{F}}}$, where $\|\cdot\|_{\mathrm{F}}$ stands for the *Frobenius norm*, T is a tree for a given graph G and id is an identity function (see: our notation from Sec. 1). It quantifies how closely the distance matrix of G is approximated by the $f$-distance matrix of T. Computing $\epsilon$ is expensive and our training does not rely on it. Our empirical results show that the relative error, $\epsilon$, can be substantially improved by using the light-weight MSE training loss (defined in Eq. 6).

We report the evaluation error for these experiments in Fig. 6 (with additional results in Fig. 8 in the Appendix). We observe that a rational function with quadratic numerator and denominator provides strong performance across different graphs. We notice that increasing the training set to $> 100$ data points does not have a substantial impact on the final error. Estimating the coefficients of $f$ provides approximation improvements across all graphs in as few as **40 training steps**.

These above results show that tree-based estimators are expressive enough to emulate integration on arbitrary graphs. This expressive power can be further enhanced by pairing them with "nonlinear" functions $f$. Thus, they explain why the presented techniques are relevant for general graphs.

## 4.4 Large Scale Transformer Experiments using FTFI

For large-scale applications of FTFI, we select Topological Vision Transformers (TopViT), [Choromanski et al., 2022], and leverage it for efficient incorporation of masking within ViTs. We provide detailed description of masked Transformers in Appendix C.

**Topological Vision Transformers with trees :** We propose an extension to TopViT that seamlessly integrates FTFI. In this extension, we model the mask matrix as an $f$-distance matrix (with learnable $f$) defined on the minimum spanning tree (MST) obtained from the 2D grid graph image encoding, where vertices correspond to different patches. We parameterize $f$ as $f_g^t \stackrel{\text{def}}{=} g(\sum_{i=0}^t a_t x^t)$. We use the linear attention mechanism introduced in Performers [Choromanski et al., 2021], where the attention kernel is written as: $\mathrm{K}(\mathbf{q}, \mathbf{k}) = \phi(\mathbf{q})^\top \phi(\mathbf{k})$ for a deterministic $\phi : \mathbb{R}^{d_{QK}} \to \mathbb{R}$, applied element-wise. We experiment with different values of hyperparameters $g$, $t$, $\phi$ and cross-heads parameter sharing strategies as shown in Table 1 (synced indicates that RPE-parameters are shared across different attention heads).

We run experiments on ImageNet and Places365 datasets using ViT-B/16 (see Table 1). For all the kernels, our variants beat the baselines. For $\phi(x) = x^4$, the best variant applies an exponentiated quadratic function, for which we apply Vandermonde matrices (see: discussion in Sec. 3.2.1). Our

Table 1: Performance of Topological Vision Transformers with tree-based masking. For each attention kernel, we present the results of the best variant in **bold** and Performer baselines in blue.

| | | | **ImageNet** | | | | | | | | | | | | | **Place365** | | |
| $\phi :=$RELU | | | | $\phi := x \to x^2$ | | | | $\phi := x \to x^4$ | | | | $\phi := \exp$ | | | | $\phi := \text{ReLU}$ | | | |
| synced | g | t | Acc. (%) | synced | g | t | Acc. (%) | synced | g | t | Acc. (%) | synced | g | t | Acc. (%) | synced | g | t | Acc. (%) |
|---|---|---|---|---|---|---|---|---|---|---|---|---|---|---|---|---|---|---|---|
| NA | NA | NA | 76.23 | NA | NA | NA | 75.03 | NA | NA | NA | 76.37 | NA | NA | NA | 76.76 | NA | NA | NA | 54.80 |
| ✓ | exp | 1 | 77.28 | ✓ | exp | 1 | 76.66 | ✓ | exp | 1 | 77.84 | ✗ | exp | 1 | **78.79** | ✗ | exp | 1 | 56.69 |
| ✓ | exp | 2 | 76.60 | ✓ | exp | 2 | 75.91 | ✓ | exp | 2 | 77.23 | ✗ | exp | 2 | 78.51 | ✗ | $z \to z^{-1}$ | 1 | 56.44 |
| ✗ | exp | 1 | **77.79** | ✗ | exp | 1 | **76.76** | ✗ | exp | 1 | 77.94 | ✗ | $z \to z^{-1}$ | 1 | 77.39 | ✗ | $z \to z^{-1}$ | 5 | 56.32 |
| ✗ | exp | 2 | 77.43 | ✗ | exp | 2 | 76.27 | ✗ | exp | 2 | **78.15** | ✗ | $z \to z^{-1}$ | 2 | 77.69 | ✗ | $z \to z^{-1}$ | 10 | 56.51 |

best variant across all kernels (**78.79%**) provides **2%** accuracy gains over the best baseline (**76.76%**). In the synced setting, we use only **three** extra learnable parameters per layer (shared in all attention heads across all layers) and obtain **1-1.5%** accuracy gains. In the asynced setting, we use a small set of **36** extra learnable parameters per layer (3 extra parameters per head). Overall, we observe that FTFI improves the approximation quality within Transformers with a minimal number of parameters. We provide additional discussions on the ViT results for ImageNet in Appendix D.5.1 and for Places365 in Appendix D.5.2.

Additional results on the I-Naturalist dataset, where we outperform various low-rank attention baselines, are provided in Appendix D.5.3.

**Larger Transformer models:** We scale our experiments to run on the larger ViT-L architectures and evaluate on ImageNet. In this setting, we use RPE mechanism with $g = \exp$ and $t = 1$ (that provided strong performance in previous experiments) and asynced strategy. We observe that FTFI provides **7%** accuracy improvement (see: Fig. 7).

Further results on Video Transformer (ViViT) [Arnab et al., 2021] are provided in Appendix D.6. We also provide additional experiments including Gromov-Wasserstein distance computation [Vayer et al., 2018] (see Sec. D.2), along with code pointers (Appendix D).

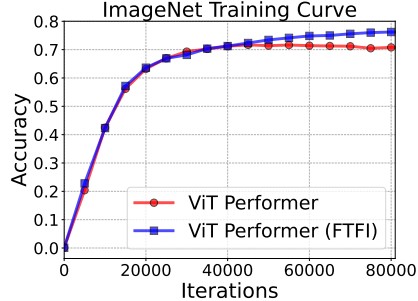

Figure 7: **Left:** Experiments with the RPE mechanism for ViT-B and on ImageNet. We observe that FTFI provides **7%** accuracy gain compared to the Performer variant.

## 5 Conclusion

We provided a new class of algorithms for fast and exact integration of tensor fields defined on weighted trees, relying on the theory of structured (in particular low displacement rank) matrices. We showed how those algorithms can be applied for accurate integration on general graphs, in particular via their minimum weight spanning trees. We presented several applications of the presented methods, from graph classification and interpolation on meshes, through graph metric approximation to Topological Vision Transformers. Our methods provide significant (5-13x) speedups while maintaining the quality of their exact counterparts.

## 6 Author Contributions

KC conceived the idea behind FTFI, proved the theoretical results, implemented FTFI algorithm and ran the vision experiments in this paper. AS integrated the FTFI algorithm in the GW style algorithms and ran some graph and point cloud classification tasks. SBRC ran graph classification experiments as well as experiments on the CUBES dataset. HL ran the experiments on the meshes. AD helped develop methods, and along with TS and SC acted as senior advisors for the project. All authors contributed to the writing of the manuscript.

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

# A Theoretical results

In this section, we provide proofs of all theoretical results in the paper.

## A.1 Proof of Lemma 3.1

*Proof.* We will apply Lemma 7.19 from [Cygan et al., 2015] (that we provide also below for reader's convenience) and its algorithmic proof. We refer to Cygan et al. [2015] for a definition of the related graph terms.

**Lemma A.1.** *Assume that* $G$ *is a graph of treewidth at most $k$, and consider a nonnegative weight function* $\mathbf{w} : V(G) \to \mathbb{R}_{\geq 0}$ *on the vertices of $G$. Then in $G$ there exists a $\frac{1}{2}$-balanced separator $X$ of size at most $k + 1$.*

Note first that for each rooted tree, we can compute the size of each of its rooted sub-trees (and store it in the root of the sub-tree) in the linear time, simply by applying dynamic programming. We can now apply the above lemma for the tree $G = \mathcal{K}$ with the weight function that assigns weight $w = 1.0$ for each vertex. By following its algorithmic proof (and using breadth first search for tree exploration), we can obtain a node $p$ and sub-trees $T_1, ..., T_l$ rooted in vertices connected with $p$, with the following properties:

- $V(T_1) \cup ... \cup V(T_l) \cup \{p\} = V(\mathcal{K})$,
- $|V(T_i)| \leq \frac{1}{2}|V(\mathcal{K})|$ for $i = 1, ..., l$ and where $||$ stands for the set size.

We then choose the first index $k$ such that $|V(T_1)| + ... + |V(T_k)| \geq \frac{3}{4}|V(\mathcal{K})|$. Note that such an index $k$ exists and $k > 1$ because of the above and the fact that our tree has at least six vertices. We define as $\mathcal{K}_{\mathrm{left}}$ a sub-tree of $\mathcal{K}$ induced by the set: $V(T_1) \cup ... V(T_{k-1}) \cup \{p\}$ and by $\mathcal{K}_{\mathrm{right}}$ a sub-tree of $\mathcal{K}$ induced by the set: $V(T_k) \cup ... V(T_l) \cup \{p\}$. Note that the triple $(\mathcal{K}_{\mathrm{left}}, \mathcal{K}_{\mathrm{right}}, p)$ satisfies the requirements of Lemma 3.1. That completes the proof. □

## A.2 Fast Approximate Tree-Field Integrators

If matrices $\mathbf{M} = [f(x_i + y_j)]_{i=1,...,a}^{j=1,...,b}$ from Sec. 3.2.1 do not support fast matrix-vector multiplication, the question arises whether fast approximate procedures can be applied.

### A.2.1 Random Fourier Features (RFFs)

Assume that the Fourier Transform (FT) of $f$ exists and denote it by $\tau : \mathbb{C} \to \mathbb{C}$. Note that $f$ is the inverse FT of $\tau$ and can be re-written as $f(z) = \int_{\mathbb{R}} \exp(2\pi \mathbf{i}\omega z)\tau(\omega)d\omega$. Therefore, the following holds:

$$f(x_i + y_j) = \int_{\mathbb{R}} \exp(2\pi \mathbf{i}\omega x_i) \exp(2\pi \mathbf{i}\omega y_j)\tau(\omega)d\omega. \tag{8}$$

We conclude that for any probabilistic distribution $\mathcal{P}$ on $\mathbb{R}$ with pdf $p$, $f(x_i + y_j)$ can be re-written as: $f(x_i + y_j) = \mathbb{E}[\mu(x_i)^\top \mu(y_j)]$, where random $\mu : \mathbb{R} \to \mathbb{R}^m$ is given as: $\mu(t)^\top = \frac{1}{\sqrt{m}} \left( \sqrt{\frac{\tau(\omega_l)}{p(\omega_l)}} \exp(2\pi \mathbf{i}\omega_l t) \right)_{l=1}^m$ for $\omega_1, ..., \omega_m \sim \mathcal{P}$ and $m \in \mathbb{N}_+$. Thus matrix $\mathbf{M}$ can be unbiasedly approximated as: $\mathbf{M} \approx \mathbf{U}\mathbf{W}^\top$ for $\mathbf{U} \in \mathbb{R}^{a \times m}$, $\mathbf{W} \in \mathbb{R}^{b \times m}$ with rows given by $\mu(x_1)^\top, ..., \mu(x_a)^\top$ and $\mu(y_1)^\top, ..., \mu(y_b)^\top$ respectively. Matrix-vector product $\mathbf{M}\mathbf{v}$ can be then unbiasedly approximated as $\mathbf{U}(\mathbf{W}^\top \mathbf{v})$ and computed in time $O((a + b)m)$. For $m \ll \frac{ab}{a+b}$, substantial computational gains are obtained. In particular, if $m = O(\log^d(a + b))$, the approximate $f$-integration is conducted in time $O(N\log^{d+1}(N))$. Note that $m$ controls estimator's variance, thus decreasing $m$ increases the error.

### A.2.2 Non-Uniform FFT (NU-FFT)

We will now propose a closely-related method, that relies on the non-uniform FFT (NU-FFT).[2]

---
[2]See [Greengard and Lee, 2004] for an excellent introduction.

Denote: $\mathbf{g} = \mathbf{M}\mathbf{v}$ for a given $\mathbf{v} = (v_1, ..., v_b)^\top \in \mathbb{R}^b$. Define: $g(x) = \int_R f(x - z)P(z)dz$, where $P$ is given as: $P(z) = \sum_{j=1}^{b} v_j \delta(z - z_j)$, and furthermore: (1) $\delta$ is a *delta-Dirac* function, (2) $z_j = -y_j$. Our goal is to efficiently evaluate function $g$ in points: $\{x_1, ..., x_a\}$.

Assume that the inverse FT of $g$ exists and denote it by $\eta : \mathbb{C} \to \mathbb{C}$. Note that $g$ is the FT of $\eta$ and can be written as: $g(x) = \int_{\mathbb{R}} \eta(\omega) \exp(-2\pi \mathbf{i}\omega x)d\omega$. Since $g$ is also a convolution of $f$ and $P$, $\eta$ is a product of the inverse FTs: $\rho$ and $R$ respectively. Therefore, we can write: $g(x) = \int_{\mathbb{R}} \rho(\omega)R(\omega) \exp(-2\pi \mathbf{i}\omega x)d\omega$, where $R(\omega) = \sum_{j=1}^{b} v_j \exp(2\pi \mathbf{i}\omega z_j)$. Now, function $g$ can be evaluated for $\{x_1, \ldots, x_a\}$ as follows: (1) a quadrature method is applied to obtain points: $\omega_1, ..., \omega_r$ (and corresponding weights) for the approximate computation of the integral defining $g$, (2) the NU-FFT is applied to compute $R(\omega)$ simultaneously in those points in polylog-linear time, (3) given pre-computed $(\rho(\omega_i)R(\omega_i))_{i=1}^{r}$ (and the quadrature weights), NU-FFT is applied again to compute quadrature-based approximation of $g$.

The $f$-integration process applying this method runs in polylog-linear time since the computation of $\mathbf{g} = \mathbf{M}\mathbf{v}$ takes polylog-linear time. A prominent application is $f$ given as: $f(x) = \frac{\sin(x)}{x}$, with $\rho$ being a renormalized indicator of belonging to interval $[-0.5, 0.5]$. In this setting, the integral defining $g$ is thus limited to $[-0.5, 0.5]$. Interestingly, for $f(x) = \frac{\sin(x)}{x}$ we can also apply methods from Sec. 3.2.1 (see: our discussion below on the trigonometric case).

### A.2.3  Additional implications of Lemma 3.3

**Products of exponentials and polynomials:**  Note that a Hadamard (element-wise) product of two outer-product matrices is itself an outer-product matrix. Using the analysis from the polynomial and exponential cases, we conclude that $\mathbf{M}$ is a sum of a constant number of terms, each being an outer-product matrix. Thus the same conclusion follows.

**The case of the trigonometric $f$:**  If $f(x) = \cos(x)$ then it can be re-written as: $f(x) = \frac{\exp(\mathbf{i}x)+\exp(-\mathbf{i}x)}{2}$. Observe that the cordiality property is preserved under linear combination of the finite number of cordial functions. We can thus conclude that analogous results as the above for $f(x) = \exp(\lambda x)$ can be derived for $f(x) = \cos(x)$. That is also the case for $f(x) = \sin(x)$ that can be re-written as: $f(x) = \frac{\exp(\mathbf{i}x)-\exp(-\mathbf{i}x)}{2\mathbf{i}}$. In both cases, we extend the domain from $\mathbb{R}$ to $\mathbb{C}$, but this does not affect the analysis.

So far we have not put any restrictions on the tree weights. If we restrict all weights to be the same (without loss of generality, equal to one), then the problem becomes easier. In this case for any function $f$, matrices $\mathbf{C}$ and $\mathbf{C}^\top$ are Hankel [Brent, 2010] (constant on each anti-diagonal and belonging to LDR class). Then, matrix-vector multiplication can be done in $O((a + b) \log(a + b))$. The analysis from the proof of Lemma 3.3 for $d = 1$ can be repeated. We conclude that $f$-integration can be conducted in $O(N \log^2(N))$ time for $N$-vertex unweighted trees and any $f : \mathbb{R} \to \mathbb{R}$. This was already proven in [Choromanski et al., 2022].

**Trees with positive rational weights:**  Assume that tree weights take values of the form: $\{\frac{e}{q} : e \in \{1, ..., p\}\}$ for some $p, q \in \mathbb{N}_+$. Then, matrices $\mathbf{C}$ and $\mathbf{C}^\top$ do not need to be Hankel, but can be embedded into Hankel matrices with rows/columns corresponding to distances $\frac{l}{q}$ from the pivot, where $l = \{0, ..., mp\}$ and $\frac{mp}{q}$ is the largest distance between a vertex and the pivot. Tensor $\mathbf{X}$ can also be padded into a larger one with extra rows/columns (corresponding to unrealized distances) set to zero. If $p$ is constant, the asymptotic time complexity remains the same as in the previous case, but the algorithm might not be practical since the number of rows and columns grows by a multiplicative factor of $p$. For certain non-cordial $f$, the algorithm can be modified for potential gains.

## B  Additional Related Work

In this section we provide additional related works. One of the methods to tackle this problem is via iterative methods [Koutis et al., 2012] like Arnoldi iteration [Arnoldi, 1951], Conjugate Gradient [Shewchuk, 1994] and the celebrated Spielman-Teng algorithm [Spielman and Teng, 2012] for symmetric diagonally dominant (SDD) matrices. There are a number of extensions and variations of the above methods [Blelloch et al., 2011, Boman et al., 2008, Christiano et al., 2010, Koutis and

Miller, 2007, Spielman and Teng, 2008, Daitch and Spielman, 2008, Koutis and Miller, 2008].They mainly take into account the structure of the matrix (SDD) [Koutis et al., 2010, 2011a, 2012], embedding of a graph into low stretch spanning trees [Elkin et al., 2005], graph sparsification [Spielman and Teng, 2010] and the choice of a good *pre-conditioner* [Maggs et al., 2003, Koutis et al., 2011b]. We want to emphasize that the research on low stretch trees for general graphs is orthogonal to the main topic of this work. In our manuscript, we show in particular how to conduct efficient integration on arbitrary trees. Thus our work can be naturally combined with those algorithms to leverage all the above low stretch tree constructions for a better approximation of the graph's metric.

The other class of method comes from the celebrated work of [Al-Mohy and Higham, 2011] and there are a number of extensions of this work [Kloster and Gleich, 2023, Al-Mohy and Higham, 2010, Moore, 2011, Moler and Van Loan, 2003, Auckenthaler et al., 2010].

Another class of methods is via sampling, where one samples a subset of a large matrix, which is then used to approximate the matrix-vector multiplication (i.e. Monte Carlo sampling) methods [Drineas et al., 2006, Drineas and Kannan, 2001, Acebron, 2019, Acebron et al., 2019, Benzi et al., 2017, Martinsson, 2019].

We note that none of these methods are directly applicable in our cases as our $f$-matrix is neither Hermitian or SDD. The randomized algorithms are harder to use in the setting of training of a neural network. Moreover our method is *exact* on *trees*, where all the above methods are approximations.

## C  Topological Transformers

---

**Algorithm 1** General Efficient Low-Rank Masked Attention from Choromanski et al. [2022]

---

**Input:** Query/key matrices: $\mathbf{Q}, \mathbf{K} \in \mathbb{R}^{L \times d_{QK}}$, value matrix $\mathbf{V} \in \mathbb{R}^{L \times d}$, mask $\mathbf{M} \in \mathbb{R}^{L \times L}$, procedure $\mathrm{FastMult}_{\mathbf{M}} : \mathbb{R}^L \to \mathbb{R}^L$ calculating $\mathbf{Mx}$ (or its approximation) for the input $\mathbf{x} \in \mathbb{R}^L$, kernel feature map: $\phi : \mathbb{R}^{d_{QK}} \to \mathbb{R}^m$. $\mathrm{vec}(\cdot)$ denotes vectorization.
**Output:** Masked low-rank attention embeddings using $\phi$.
1. Compute matrices $\mathbf{V}^1 \in \mathbb{R}^{L \times (md)}$, $\mathbf{V}^2 \in \mathbb{R}^{L \times m}$ with rows defined as: $\mathbf{V}^1_{i:} = \mathrm{vec}(\phi(\mathbf{k}_i^\top)\mathbf{v}_i)$, $\mathbf{V}^2_{i:} = \phi(\mathbf{k}_i^\top)^\top$, where $\mathbf{k}_i/\mathbf{v}_i$ stands for the ith row of $\mathbf{K}/\mathbf{V}$.
2. $\tilde{\mathbf{D}}^1 := [\mathrm{FastMult}_{\mathbf{M}}(\mathbf{V}^1_{:1}), ..., \mathrm{FastMult}_{\mathbf{M}}(\mathbf{V}^1_{:md})] \in \mathbb{R}^{L \times md}$,
$\tilde{\mathbf{D}}^2 := [\mathrm{FastMult}_{\mathbf{M}}(\mathbf{V}^2_{:1}), ..., \mathrm{FastMult}_{\mathbf{M}}(\mathbf{V}^2_{:m})] \in \mathbb{R}^{L \times m}$ for $\mathbf{V}^{1/2}_{:i}$ denoting ith column of $\mathbf{V}^{1/2}$.
3. Output the embedding $\mathbf{r}_i$ of the ith tokens as: $\mathbf{r}_i = \frac{\phi(\mathbf{q}_i^\top)^\top \mathrm{devec}(\tilde{\mathbf{D}}^1_{i:})}{\phi(\mathbf{q}_i^\top)^\top (\tilde{\mathbf{D}}^2_{i:})^\top}$, where $\mathbf{q}_i$ is the ith row of $\mathbf{Q}$ and $\mathrm{devec}(\cdot)$ devectorizes its input back to $\mathbb{R}^{m \times d}$.

---

We now recall the formulation of general masked transformers.

Let us denote by $L$ the number of input tokens. The attention used in a regular Transformer linearly projects their representations into three learnable matrices $\mathbf{Q}, \mathbf{K} \in \mathbb{R}^{L \times d_{QK}}$, $\mathbf{V} \in \mathbb{R}^{L \times d}$ called *queries*, *keys* and *values* respectively.

**Definition C.1** (general masked attention). *General masked attention* is given by the following equation, where $\mathbf{M} \in \mathbb{R}^{L \times L}$ is the *mask matrix*, and $\mathbf{A} \in \mathbb{R}^{L \times L}$ is the so-called *masked attention matrix* (MAM): which is defined as:

$$\mathrm{Att}_{\mathrm{K}}(\mathbf{Q}, \mathbf{K}, \mathbf{V}, \mathbf{M}) = \mathbf{D}^{-1} \mathbf{A} \mathbf{V},$$
$$\mathbf{A} = \mathbf{M} \odot \mathrm{K}(\mathbf{Q}, \mathbf{K}), \quad \mathbf{D} = \mathrm{diag}(\mathbf{A} \mathbf{1}_L),$$

(9)

where $\odot$ denotes the element-wise (Hadamard) matrix product, $\mathrm{K} : \mathbb{R}^d \times \mathbb{R}^d \to \mathbb{R}$ is some kernel function and $\mathrm{K}(\mathbf{Q}, \mathbf{K})$ is a kernel matrix defined as: $\mathrm{K}(\mathbf{Q}, \mathbf{K})_{i,j} = \mathrm{K}(\mathbf{q}_i^\top, \mathbf{k}_j^\top)$ for the $ith$ row $\mathbf{q}_i$ of $\mathbf{Q}$ and the jth row $\mathbf{k}_j$ of $\mathbf{K}$ respectively. We call $\mathbf{A}' = \mathrm{K}(\mathbf{Q}, \mathbf{K})$ the unmasked attention matrix (UAM). Note that when $\mathrm{K}$ is the softmax function, we recover the well-known attention mechanism in Transformers.

Here $\mathbf{1}_L$ is the all-ones vector of length $L$, and $\mathrm{diag}(\cdot)$ is a diagonal matrix with the input vector as the diagonal. The time complexity of computing (9) is $O(L^2 d)$.

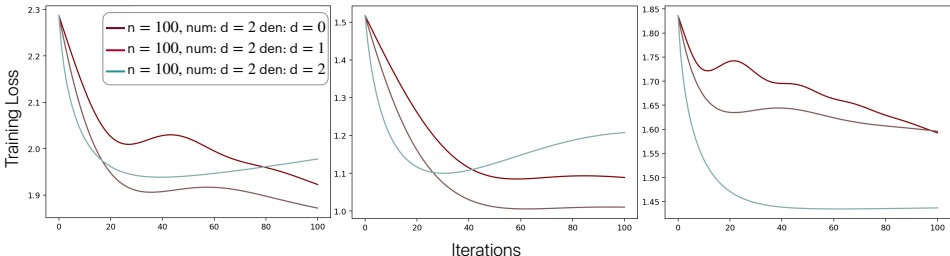

Figure 8: Relative Frobenius norm error as a function of the number of training iterations for different sizes $n$ and learnable quadratic $f$. We report the results for 3 mesh graphs from Thingi10k.

If the kernel K admits (at least in expectation) a dot-product decomposition, i.e. $K(\mathbf{x}, \mathbf{y}) = \mathbb{E}[\phi(\mathbf{x})^\top \phi(\mathbf{y})]$ for some mapping: $\phi : \mathbb{R}^{d_{QK}} \to \mathbb{R}^m$ (and some $m > 0$). $\phi(\mathbf{u})$ is called a *(random) feature map* (RFM) for $\mathbf{u} \in \mathbb{R}^d$. For $\mathbf{Q}', \mathbf{K}' \in \mathbb{R}^{L \times m}$ with rows given as $\phi(\mathbf{q}_i^\top)^\top$ and $\phi(\mathbf{k}_i^\top)^\top$ respectively, RFM-based kernel linearization leads directly to the efficient unmasked attention mechanism of the form:

$$\widehat{\mathrm{Att_K}}(\mathbf{Q}, \mathbf{K}, \mathbf{V}) = \widehat{\mathbf{D}}^{-1}(\mathbf{Q}'((\mathbf{K}')^\top \mathbf{V})),$$
$$\widehat{\mathbf{D}} = \mathrm{diag}(\mathbf{Q}'((\mathbf{K}')^\top \mathbf{1}_L)). \tag{10}$$

Here $\widehat{\mathrm{Att_K}}$ stands for the approximate attention and brackets indicate the order of computations. Such a mechanism is characterized by time complexity $O(Lmd)$ as opposed to $O(L^2 d)$ for regular attention. If $m \ll L$, computational gains are obtained.

The central question in [Choromanski et al., 2022] was how to incorporate the masking in the linear attention as above. Note that in this case $\mathbf{A}'$ is never materialized. Building on the work of [Luo et al., 2021], the authors [Choromanski et al., 2022] propose a general algorithm that efficiently implements masked linear attention.

In this work, we use different mappings $\phi$ (see Table 1). Our key contribution in this work is to propose a novel mask matrix $\mathbf{M}$ and the implementation of a fast matrix multiplication by $\mathbf{M}$. The above result then allows us to construct novel classes of Topological Transformers.

## D   Experimental Details and Additional Experiments

In this section, we provide additional details regarding the experimental setup and present additional results from our experiments. Our code is available at `https://github.com/brcsomnath/FastTreeIntegrator`. Specifically, we provide there the code for: (1) our algorithm leveraging IntegratorTree data structure (depicted in Fig 1), (2) adaptation to the Gromov-Wasserstein-type computation, (3) graph classification and (4) experiments on interpolation on meshes.

### D.1   Additional experiments for graph metric approximation with $f$-distance matrices

We present additional results for the training loss, relative Frobenius Norm Error ($\epsilon$), for more samples from the Thingi10K dataset (to complement the results in Fig. 6). In Fig. 9, we observe that in most cases having rational functions with higher polynomial degrees results in lower training loss.

We also perform similar experiments for graph classification on the CUBES dataset Hanocka et al. [2019]. Specifically, we investigate how the polynomial degree affects the graph classification performance in Fig. 9 (left). We observe that increasing the polynomial degree improves the classification accuracy up to a certain degree. For the same dataset, we also compute the training loss for different polynomial degrees in Fig. 9 (right). Similarly, we observe that higher-degree rational functions achieve lower training loss for fitting the polynomial coefficients.

Moreover, we benchmark FTFI on ModelNet10 [Wu et al., 2015], a dataset for 3D Point Cloud (PC) classification. For each PC, we create an $\epsilon$-neighborhood-graph and use FTFI for graph classification The Shortest Path kernel achieves an accuracy of $39.6\%$, whereas our FTFI with the degree-2

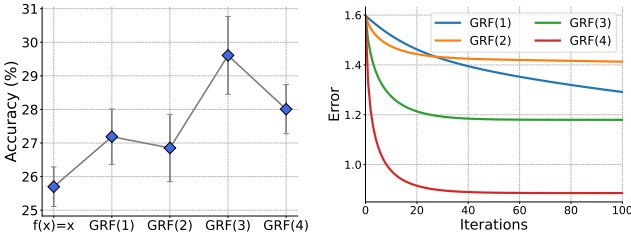

Figure 9: **Left**: Variation in FTFI performance with different $f$-distance functions on the CUBES dataset. We use general rational functions (GRF) of varying polynomial degrees. GRF($i$) indicates a rational function of the $i$-th degree. We observe a general trend of accuracy increase with function complexity up to a certain degree. The coefficients of the GRF were learnt using a few graph instances. **Right**: We show the training loss curves for estimating the coefficients of the rational function, $f$, for samples in the CUBES dataset. We report the training loss for rational functions with varying polynomial degrees. We observe that the training loss is lower when we use rational functions with high-degree polynomials.

polynomial improves the accuracy to $44.2\%$ ($10\%$ relative improvement over the baseline), similar to the observation in 9.

## D.2 Integration of FTFI into GW-style algorithms

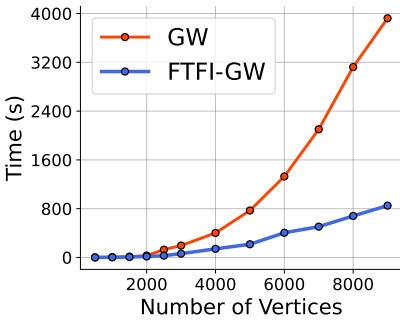

Figure 10: Comparison of field integration time between GW and FTFI-GW. We observe that FTFI achieves significant computation time gain over the baseline.

Wasserstein distance has found many uses in ML, particularly due to it's principled approach to compare probability distributions. Gromov Wasserstein Mémoli [2011] discrepancy is an extension of Wasserstein distance to graph structured data, with a lot of downstream applications like graph clustering and classification. Inspired by the work of [Choromanski et al., 2023], we follow the exact same procedure in the integration of FTFI in the conditional gradient algorithm. The FTFI can be injected seamlessly in place of the Fast Matrix Multiplication (FMM) algorithms in Algorithm 2 and Algorithm 3 (see [Choromanski et al., 2023]).

Our method GW-FTFI run consistently 2-6x faster than the baseline methods using the Shortest Path kernel, with *no drop* in accuracy in computing the associated costs (Figure 10). The plots shown are obtained by averaging over 10 seeds and random trees of various sizes. For the baseline experiments, we use the implementation from the POT library [Flamary et al., 2021].

## D.3 Interpolation on Meshes

In this section, we present implementation details for the mesh interpolation experiments in Section 4.2. All experiments were run on a computer with an i9-12900k CPU and 64GB memory.

In the vertex normal prediction task in Section 4.2, we choose 40 meshes for 3D-printed objects with a wide range of size from the Thingi10K [Zhou and Jacobson, 2016] dataset with the File IDs:

Table 2: Statistics of the graph classification datasets used in this paper.

| DATASETS | # Graphs | # Labels | Avg. # Nodes | Avg. # Edges | # Node Labels | # Node Attributes |
|---|---|---|---|---|---|---|
| MUTAG | 188 | 2 | 17.93 | 19.79 | 7 | - |
| PTC-MR | 344 | 2 | 14.29 | 14.69 | 19 | - |
| ENZYMES | 600 | 6 | 32.63 | 62.14 | 3 | 18 |
| PROTEINS | 1113 | 2 | 39.06 | 72.82 | 3 | 1 |
| D&D | 1178 | 2 | 284.32 | 715.66 | 82 | - |
| IMDB BINARY | 1000 | 2 | 19.77 | 96.53 | - | - |
| IMDB MULTI | 1500 | 3 | 13.0 | 65.94 | - | - |
| NCI1 | 4110 | 2 | 29.87 | 32.30 | 37 | - |
| COLLAB | 5000 | 3 | 74.49 | 2457.78 | - | - |
| REDDIT BINARY | 2000 | 2 | 429.63 | 497.75 | - | - |
| REDDIT MULTI-5K | 4999 | 5 | 508.52 | 594.87 | - | - |
| REDDIT MULTI-12K | 11929 | 11 | 391.41 | 456.89 | - | - |

Table 3: Feature processing time of FTFI compared to exact shortest path kernel computation. We observe that FTFI achieves significant speedups up to 90% reduction in processing time. All times are reported in seconds (s).

| | DATASETS | | | | | |
|---|---|---|---|---|---|---|
| Algorithm | IMDB BINARY | IMDB MULTI | REDDIT BINARY | REDDIT MULTI-5K | REDDIT MULTI-12K | COLLAB |
| BGFI | 5.6 | 7.6 | 3371.9 | 6267.6 | 8086.3 | 209.1 |
| FTFI | 4.3 | 4.7 | 338.2 | 755.3 | 1959.5 | 232.4 |
| Improvement | +23.2% | +38.2% | +90.0% | +88.0% | +75.8% | -11.1% |

| | DATASETS | | | | | |
|---|---|---|---|---|---|---|
| Algorithm | MUTAG | ENZYMES | NCI1 | PTC-MR | D&D | PROTEINS |
| BGFI | 0.88 | 3.68 | 32.8 | 0.89 | 715.4 | 14.9 |
| FTFI | 0.37 | 4.39 | 20.2 | 0.93 | 325.3 | 18.6 |
| Improvement | +58.0% | -19.3% | +38.4% | -4.5% | +54.5% | -24.8% |

[60246, 85580, 40179, 964933, 1624039, 91657, 79183, 82407, 40172, 65414, 90431, 74449, 73464, 230349, 40171, 61193, 77938, 375276, 39463, 110793, 368622, 37326, 42435, 1514901, 65282, 116878, 550964, 409624, 101902, 73410, 87602, 255172, 98480, 57140, 285606, 96123, 203289, 87601, 409629, 37384, 57084]

For both our FTFI and the baseline BFFI methods, we do a grid-search over the hyper-parameter $\lambda$ for each mesh and report the pre-processing time associated with the hyper-parameter(s) that give(s) us the best cosine similarity.

## D.4 Additional Details on Graph Classification

We conduct graph classification experiments on a wide range of benchmark datasets. We report the dataset statistics for the graph classification datasets in Table 2. More details about the datasets are available in Morris et al. [2020]. To evaluate the performance of the different kernels, we employ the framework proposed by [Errica et al., 2020]. In particular, 10-fold cross-validation is used to obtain an estimate of the generalization performance of our method and the baseline method. We repeat this cross validation experiment 5 times to get a robust estimation and report the standard deviation for each setup.

To obtain graph features, we follow the approach presented in [de Lara and Pineau, 2018]. In this setting, we obtain the $k$-smallest eigenvalues from the approximated kernel from FTFI and forward these features to a random forest classifier for classification. For BGFI, we perform the same process obtaining the $k$-smallest eigenvalues from the exact shortest kernel. FTFI achieves similar performance to the BGFI while being significantly faster. We tune the hyperparameter $k$ independently for each method.

In Table 4, we report the results for a wide range of baselines and compare FTFI. We observe that FTFI achieves competitive performance among various strong kernel-based classification baseline approaches. Note that FTFI results are not directly comparable with other approaches, as FTFI constructs an intra-graph kernel while other methods use inter-graph kernels. Despite the aforementioned considerations, we contend that positioning our results within the broader framework of alternative methodologies demonstrates that FTFI remains a compelling approach, owing to its speed and comparable classification accuracy.

## D.5 Additional details on experiments for Topological transformers

In this subsection, we provide additional training details for our image classification tasks. Table 5 and table 6 present the architectural as well as the training details.

We train the ViT models starting from their pretrained checkpoint (pretrained on ImageNet-21k). We replace the dense attention in ViT by the Performer attention (see Equation 10). We use Algorithm 1 to efficiently incorporate the mask matrix $\mathbf{M}$ in the attention mechanism.

### D.5.1 ImageNet

We have already provided comparison with SOTA efficient-attention methods: low-rank attention Transformers in Sec 4.4, quality-wise. On standard ImageNet benchmark, our best Transformer with FTFI provide 78.15% accuracy, as compared to 76.37% of the best low-rank -attention variant (obtained by testing three different linear variants). That gives 1.78% accuracy improvement with only 3 extra trainable parameters per head (36 extra trainable parameters per layer). We have also run the experiments with cosFormer. It achieved 76.3% accuracy (consistent with what is reported in the literature, see [8]), lower than both: our method and the best tested low-rank attention variant. The RF-Gate-Gaussian achieved 76.35% accuracy, which is is still lower than both: FTFI and the best tested low-rank attention variant.

### D.5.2 Places365

We have also conducted tests on another challenging dataset: Places365. In the paper, we report 1.71% accuracy improvement over low-rank attention Transformer (56.51% accuracy vs 54.8% accuracy). For the rebuttal, we also run the experiment with cosFormer which achieved 55.4% accuracy (consistent with what is reported in the literature, see: [8]). This is still 0.93% behind our method. The RF-Gate-Gaussian achieved accuracy 55.1%, lower than this of cosFormer.

### D.5.3 I-naturalist 2017

I-naturalist is yet another challenging dataset, with 10K classes, diverse image quality and significant class imbalance. Transformer with FTFI provides 1% accuracy improvement over its regular low-rank attention counterpart and the cosFormer. Furthermore, FTFI achieved 0.8% improvement over RF-Gate-Gaussian. The convergence of the FTFI variant is 20-23% faster than this of its regular low-rank attention counterpart, the cosFormer and RF-Gate-Gaussian.

## D.6 Video Vision Transformer

ViViT ([Arnab et al., 2021]) is a novel architecture that adapts the Vision Transformer (ViT) for video processing. It efficiently handles the spatiotemporal dimensions of video data by factorizing the input and applying attention mechanisms across both space and time. This allows ViViT to capture complex motion patterns and long-range dependencies in videos.

Applying FTFI with a topological masking mechanism to the ViViT architecture (factorized Transformer model variant, trained from scratch, as described in Arnab et al. [2021]) results in a **0.8%** absolute improvement on the Kinetics dataset ([Kay et al., 2017]). The experimental setup follows Arnab et al. [2021]. To the best of our knowledge, this is the first application of Topological Transformers to video data.

Table 4: Comparison of FTFI with a broad range of graph kernel-based classification approaches. We observe that FTFI achieves performance similar to that of Exact SP, its exact counterpart, across almost all datasets. The baseline results have been compiled from Nikolentzos et al. [2021]. OOT and OOM indicate that the corresponding algorithm ran out of time or memory respectively.

| | DATASETS | | | | | |
|---|---|---|---|---|---|---|
| Algorithm | IMDB BINARY | IMDB MULTI | REDDIT BINARY | REDDIT MULTI-5K | REDDIT MULTI-12K | COLLAB |
| VH | 50.0 (± 0.0) | 33.3 (± 0.0) | 50.0 (± 0.0) | 20.0 (± 0.0) | 21.7 (± 1.5) | 52.0 (± 0.1) |
| RW | 64.1 (± 4.5) | 44.6 (± 4.1) | OOT | OOT | OOT | 68.0 (± 1.7) |
| SP | 58.2 (± 4.7) | 39.2 (± 2.3) | 81.7 (± 2.5) | 47.9 (± 1.9) | OOT | 58.8 (± 1.2) |
| GR | 66.1 (± 2.7) | 39.5 (± 2.7) | 76.1 (± 2.6) | 34.7 (± 2.0) | 23.0 (± 1.4) | 73.0 (± 2.0) |
| WL-VH | 70.7 (± 6.8) | 51.3 (± 4.4) | 67.8 (± 3.5) | 50.5 (± 1.6) | 38.7 (± 1.7) | 78.3 (± 2.1) |
| WL-SP | 58.2 (± 4.7) | 39.2 (± 2.3) | OOT | OOT | OOT | 58.8 (± 1.2) |
| WL-PM | 73.6 (± 3.4) | 49.1 (± 5.5) | OOM | OOM | OOM | OOM |
| WL-OA | 72.6 (± 5.5) | 51.1 (± 4.3) | 89.0 (± 1.3) | 54.0 (± 1.2) | OOT | 80.5 (± 2.0) |
| NH | 71.6 (± 4.5) | 50.5 (± 5.0) | 81.2 (± 2.0) | 49.9 (± 2.4) | 39.6 (± 1.4) | 81.1 (± 2.4) |
| NSPDK | 67.4 (± 3.3) | 44.6 (± 3.8) | OOT | OOT | OOT | OOT |
| Lo-$\vartheta$ | 51.0 (± 4.2) | 39.8 (± 2.6) | OOT | OOT | OOT | OOT |
| SVM-$\vartheta$ | 52.3 (± 4.0) | 39.5 (± 2.7) | 74.8 (± 2.6) | 31.4 (± 1.1) | 22.9 (± 0.9) | 52.0 (± 0.1) |
| ODD-STh | 65.0 (± 4.0) | 46.7 (± 3.4) | 52.1 (± 3.2) | 43.1 (± 1.8) | 30.0 (± 1.6) | 52.0 (± 0.1) |
| PM | 66.3 (± 4.2) | 46.1 (± 3.8) | 86.5 (± 2.1) | 48.3 (± 2.5) | 41.1 (± 0.6) | 74.0 (± 2.4) |
| GH | 59.4 (± 3.4) | 39.5 (± 2.6) | OOT | OOT | OOT | 60.0 (± 1.4) |
| SM | OOT | OOT | OOM | OOM | OOM | OOT |
| PK | 51.7 (± 3.7) | 34.5 (± 3.0) | 63.9 (± 3.0) | 34.9 (± 1.7) | 23.9 (± 1.2) | 57.0 (± 1.2) |
| ML | 69.9 (± 4.8) | 47.7 (± 3.2) | 89.4 (± 2.1) | 35.4 (± 2.0) | OOM | 75.6 (± 1.6) |
| CORE-WL-VH | 73.5 (± 6.1) | 51.7 (± 4.1) | 73.0 (± 4.5) | 51.1 (± 1.6) | 40.2 (± 1.8) | 84.5 (± 2.0) |
| CORE-SP | 68.5 (± 3.9) | 51.0 (± 3.5) | 91.0 (± 1.8) | OOT | OOM | OOT |
| FTFI | 65.1 (± 1.6) | 46.4 (± 1.9) | 83.7 (± 1.3) | 43.8 (± 2.0) | 31.8 (± 0.3) | 63.7 (± 0.3) |
| BGFI | 65.1 (± 2.0) | 47.6 (± 2.0) | 84.3 (± 3.5) | 44.0 (± 1.9) | 37.6 (± 0.3) | 75.5 (± 0.3) |

| | DATASETS | | | | | |
|---|---|---|---|---|---|---|
| Algorithm | MUTAG | ENZYMES | NCI1 | PTC-MR | D&D | PROTEINS |
| VH | 69.1 (± 4.1) | 20.0 (± 4.8) | 55.7 (± 2.0) | 57.1 (± 9.6) | 74.8 (± 3.7) | 71.1 (± 4.4) |
| RW | 81.4 (± 8.9) | 16.7 (± 1.8) | OOT | 54.4 (± 9.8) | OOM | 69.5 (± 5.1) |
| SP | 82.4 (± 5.5) | 37.3 (± 8.7) | 72.5 (± 2.0) | 60.2 (± 9.4) | 77.9 (± 4.5) | 74.9 (± 3.6) |
| WL-VH | 86.7 (± 7.3) | 50.7 (± 7.3) | 85.2 (± 2.2) | 64.9 (± 6.4) | 78.7 (± 2.3) | 76.2 (± 3.5) |
| WL-SP | 81.4 (± 8.7) | 27.3 (± 7.4) | 60.8 (± 2.4) | 54.5 (± 9.8) | 76.0 (± 3.5) | 72.1 (± 3.1) |
| WL-PM | 88.3 (± 7.1) | 57.5 (± 6.8) | 85.6 (± 1.7) | 65.1 (± 7.5) | OOM | 75.9 (± 3.8) |
| WL-OA | 87.2 (± 5.4) | 58.0 (± 5.0) | 86.3 (± 1.6) | 65.7 (± 9.6) | 77.6 (± 3.0) | 76.2 (± 3.9) |
| NH | 88.3 (± 6.3) | 54.5 (± 3.6) | 84.7 (± 1.9) | 63.4 (± 9.2) | 74.6 (± 3.5) | 75.0 (± 4.2) |
| NSPDK | 85.6 (± 8.9) | 42.2 (± 8.0) | 74.3 (± 2.1) | 59.1 (± 7.3) | 78.9 (± 4.7) | 72.5 (± 2.9) |
| ODD-STh | 80.4 (± 8.8) | 32.3 (± 4.8) | 75.2 (± 2.0) | 59.4 (± 9.8) | 76.4 (± 4.5) | 70.9 (± 4.1) |
| PM | 85.1 (± 5.8) | 43.2 (± 5.3) | 73.5 (± 1.9) | 60.2 (± 8.2) | 77.9 (± 3.7) | 70.9 (± 4.4) |
| GH | 82.5 (± 5.8) | 37.2 (± 6.6) | 71.0 (± 2.3) | 60.2 (± 9.4) | OOT | 74.8 (± 2.4) |
| SM | 85.7 (± 5.8) | 35.7 (± 5.5) | OOT | 60.2 (± 6.8) | OOM | OOM |
| PK | 76.6 (± 5.2) | 44.0 (± 6.3) | 82.1 (± 2.1) | 65.1 (± 5.6) | 77.7 (± 4.2) | 73.1 (± 4.7) |
| ML | 87.2 (± 7.5) | 48.5 (± 7.8) | 79.7 (± 1.8) | 64.5 (± 5.8) | 78.6 (± 4.0) | 74.2 (± 4.4) |
| CORE-WL-VH | 85.6 (± 6.5) | 51.7 (± 7.0) | 85.2 (± 2.2) | 65.5 (± 5.6) | 79.5 (± 3.2) | 76.5 (± 4.4) |
| CORE-SP | 85.1 (± 6.8) | 39.5 (± 9.3) | 73.8 (± 1.4) | 57.3 (± 9.7) | 79.3 (± 3.8) | 76.5 (± 3.9) |
| FTFI | 81.6 (± 3.8) | 34.6 (± 1.0) | 72.8 (± 1.2) | 60.6 (± 2.1) | 73.6 (± 2.1) | 72.5 (± 1.2) |
| BGFI | 82.2 (± 2.8) | 42.5 (± 1.8) | 73.7 (± 1.2) | 58.7 (± 2.5) | 74.8 (± 2.1) | 71.7 (± 2.0) |

Table 5: Hyperparameters for the different ViT models used in this paper

| Model | Heads | Layers | Hidden Dim. | MLP Dim. | Params | Patch Size |
|-------|-------|--------|-------------|----------|--------|------------|
| ViT-Base | 12 | 12 | 768 | 3072 | 86M | 16 |
| ViT-Large (16) | 24 | 16 | 1024 | 4096 | 307M | 16 |

Table 6: Hyperparameters for Topological Transformer experiments

| Parameter | Value |
|-----------|-------|
| Activation layer | gelu |
| Dropout prob | 0.1 |
| Attention dropout prob | 0.1 |
| Optimizer | Adam |
| Learning rate | $10^{-3}$ |
| Batch Size | 4096 |
| Compute resources | $8 \times 8$ TPUv3 |
| Number of Epochs | 300 |
| Warmup | 10K |
| weight decay | 0.1 |
| learning schedule | cosine decay |

## E  Broader Impact

We do believe that the potential impact of this work is significant, as providing both: (a) theoretical advancements in structural graph theory as well as (b) practical applications in (1) designing computationally efficient Transformers leveraging topological inductive priors, (2) graph classification and (3) interpolation on manifolds. The core problem of fast multiplication with $f$-distance matrices plays an important role in various fields: physical sciences, chemistry, and network sciences. Our main contributions are algorithmic, with no clear negative side effects. While used in the context of Transformers, they should be though applied cautiously due to the nontrivial carbon emission footprint associated with training large Transformer models.

## F  Limitations

Currently, FTFI can be applied on general graphs via certain classes of trees defined on these graphs (e.g. spanning trees), with low-distortion trees being more preferable. It would be interesting to see whether the main concepts used in the FTFI algorithm (such as the theory of balanced separators) can be directly incorporated into efficient and exact algorithms operating on general graphs (or general sparse graphs that appear in most machine learning applications). Determining general conditions on the classes of graphs and functions $f$ under consideration that are sufficient for exact sub-quadratic time integration is yet another important problem for future work.

