# OpenReview forum: "Fast Tree-Field Integrators: From Low Displacement Rank to Topological Transformers"
_NeurIPS.cc/2024/Conference — NeurIPS 2024 poster_

### Official Review · Reviewer_H9MT · 2024-07-14

**Soundness:** 3
**Presentation:** 4
**Contribution:** 3
**Rating:** 7
**Confidence:** 2

**Summary:**

In "Fast Tree-Field Integrators: From Low Displacement Rank to Topological Transformers", the authors propose a novel method for integrating scalar fields, and more generally tensor fields, on trees. Given an input tree and potentially multiple tensor fields, FTFI constructs an auxiliary binary "IntegratorTree" with nodes corresponding to sub-trees of the input tree. The efficiency of FTFIs is then demonstrated on a variety of experiments and tasks on synthetic and real-world datasets.

**Strengths:**

1. FTFI significantly improves the runtime in contrast to previous tree integration algorithms
2. The paper conducts extensive experiments, showing promising applications with Topological Vision Transformers.
3. The illustrations (Figure 1&2) are very well-made and illustrate the relevant concepts.

**Weaknesses:**

1. Figure 5 claims that FTFI achieves similar accuracy as its brute force counterpart BGFI. However, BGFI seems to outperform FTFI on almost all datasets and often with accuracy gains of >1%.
2. Many graphs in practice are not trees. The paper does not discuss any theoretical guarantees or qualitative patterns on how well FTFI can be used as approximate integrators for almost tree-like and general graphs.
3. In the conclusion, speedups of 5--13x while maintaining quality are claimed. Both of these claims don't seem to be supported by all experiments and should be framed more cautiously.

**Questions:**

In the introduction and related work, you talk about general graphs as an input (line 84), however, in section 3.1 you explicitly talk about the input tree. How do you turn the general graph into a tree? What information or what accuracy is lost in that process? Are there theoretical considerations that quantify this?

Thank you in advance for your answers!

**Limitations:**

The authors discuss limitations in the appendix.

---

> ### Author Rebuttal · Authors · 2024-08-06
>
> **General comment:**
>
> We would like to sincerely thank the Reviewer for the feedback. **We provide responses below and in the official comment titled: "Additional responses for Reviewer H9MT**.
>
> **Fig. 5: FTFI vs BGFI accuracy-wise:**
>
> Thank you very much for the comment. Given the substantial speedups, often of the order of **1.5-2x**, we consider loss of accuracy of the order of **1%** small. Following Reviewer’s suggestion, in the camera-ready version we will avoid less precise phrases to describe the results in Fig. 5 (e.g. “similar accuracy”) and simply provide all the numbers describing accuracy-speed trade-off also in the comments. We also want to notice that for a fixed budget of training time corresponding to FTFI, the BGFI results accuracy-wise were much lower than those of FTFI. We will clarify it in the final version of the paper.
>
> **5--13x speedup claims in the conclusion:**
>
> Thank you very much for the comment ! The 5--13x speedups were obtained for examples from Sec. 4.1 (Fig. 3). Those included both: synthetic networks as well as mesh-graphs of real objects. In that section, we could test the limits of our methods and consider graphs as large as **20K** nodes. The problem with using graphs of those sizes in downstream applications, where we compared different method accuracy-wise is that graphs with 15K+ nodes are infeasible for standard algorithms. Thus we needed to scale down graph sizes in those experiments and that of course made computational improvements smaller, yet still very significant.  For instance, for mesh-modeling (Fig. 4) we obtained **3x+** speedups over brute-force method and **2x+** speedups as compared with all other tested methods. For graph classification (Fig. 5), we obtained **1.5x-2x** speedups for many graphs. For Gromov-Wasserstein (Appendix D.2), we obtained up to **5x** speedups.
>
> The exact statement in the conclusion section is: “Our methods provide significant (5-13x) speedups while maintaining the quality of their **exact counterparts**”. The “exact counterparts”-phrase is very important here. We do not claim that across all the experiments, we maintain the quality of previous more brute-force approaches. We claim that across all the experiments we maintain the quality of methods that use trees and apply brute-force integration on them (the exact counterparts). This is true, since our efficient algorithms is **numerically equivalent** to the one conducting brute-force integration on trees.
> We will clarify it in the final version of the paper, by explaining what we mean by “exact counterparts”.
>
> **FTFI to approximate almost-tree like graphs and general graphs: theoretical analysis (PART I: SECOND PART IN THE EXTRA OFFICIAL COMMENT, PARAGRAPH: THEORETICAL GUARANTEES FOR  ALMOST-TREE LIKE GRAPHS):**
>
> Thank you for an excellent question. **We provide the first part of the response here and the second part (where we explicitly provide guarantees for almost-tree like graphs in the official comment, title: "Theoretical guarantees for almost-tree like graphs").**
> The general answer to the question of the quality guarantees of FTFI is that those are the derivatives of the guarantees of the distortion ratio of the underlying trees. The quality of the general graph metric approximation with tree-metrics is a research area on its own, with voluminous literature. We provided a summary in Sec. 2 as well as in Appendix B. Notably, tree-metrics is a well-established tool, used in several applications, e.g. in distributed & online algorithms and biology (see: [1,2,3]). Since in the presentation of the FTFI algorithm we did not focus on any particular tree construction, we did not provide corresponding theoretical analysis. However, since in the experiments we focus on minimum spanning trees (MSTs), as particularly easy to construct, we would like to note that the so-called near minimum spanning trees, often provide **constant** average distortion ([4]). In the camera-ready version, following Reviewer’s suggestions. We will discuss this construction and the corresponding theoretical results in more depth, as particularly relevant for us.
>
> We want to emphasize that standard algorithmic distortion upper bounds on tree-metrics provide only **very loose** upper bounds for the distortion of the FTFI since our algorithm is applied in the ML setting, where the nonlinear mapping f is learned. Let us explain it in more depth. Assume that we want to approximate h(d), where d is the shortest path distance between two nodes: x and y in the original graph and h is some function. Assume also that in order to do it, we apply tree T. Assume that in that tree the distance between x and y is l. Then even if the distortion is large, i.e. l>>d, as long as we can find a function f such that
> f(l) accurately approximates h(d), the approximation quality will be good. All that is needed in addition to that is to make sure that f can be parameterized in such a way that it belongs to the considered by us class of cordial functions. We very explicitly show that this strategy works well in Sec. 4.3, where we take: h=identity and effectively demonstrate that learnable cordial f can achieve distortion much lower that theoretical log(N) ratio (for the worst-case optimal Fakcharoenphol trees ([5])). In that section, the log(N) distortion would translate to losses >=9, whereas losses achieved by us there are <= 1 or <=1.3.
>
> [1] Efficient distributed approximation algorithms via probabilistic tree embeddings, Khan et al.,
>      PODC 2008.
>
> [2] K-server via multiscale entropic regularization, Bubeck et al., STOC 2018
>
> [3] Distorted metrics on trees and phylogenetic forests, Mossel et al., IEEE ACM Trans. Comput.
>      Biol. Bioinform, 2007.
>
> [4] On notions of distortion and an almost minimum spanning tree with constant average distortion., Bartal et al., STOC 2016.
>
> [5] A tight bound on approximating arbitrary metrics by tree metrics, Fakcharoenphol et al., . J. Comput. Syst. Sci.

---

> > ### Author Response · Authors · 2024-08-09
> > **Addressing comments of Reviewer H9MT**
> >
> > Dear Reviewer H9MT,
> >
> > We would like to once more sincerely thank you for all the comments and very useful feedback. We think that we have addressed in depth all Reviewer's questions. Please let us know. If the Reviewer has any additional questions, we would be more than happy to answer them.
> >
> > Yours sincerely,
> >
> > The Authors

---

> > > ### Author Response · Authors · 2024-08-11
> > >
> > > Dear Reviewer H9MT,
> > >
> > > We would like to once more sincerely apologize for taking your time. As we mentioned before, we believe we have addressed all Reviewer’s comments. We do hope that the Reviewer can update the score correspondingly. If the Reviewer has any additional questions, please let us know and we will be happy to address them. Thank you very much !
> > >
> > > Yours sincerely,
> > >
> > > The Authors

---

> > ### Comment · Reviewer_H9MT · 2024-08-12
> >
> > Thank you very much for your in-depth response and your proposed clarifications! I especially find the part on the almost-tree-likeness very interesting. I have no further questions.
> >
> > I will raise my score as I believe this to be a very good paper. However, I am interested to learn of the opinion of the other reviewers and the AC, particularly from those with a stronger background in this topic.

---

### Official Review · Reviewer_o1jE · 2024-07-16

**Soundness:** 4
**Presentation:** 2
**Contribution:** 3
**Rating:** 7
**Confidence:** 2

**Summary:**

The paper tackles the problem of integrating tensor fields defined on graphs. The paper suggests Fast Tree-Field Integrators (FTFI) that integrate tensors on weighted trees with reduced time complexity, which is based on their data structure called IntegralTrees (IT). The idea is original and new, and the algorithm is carefully designed. All the properties of Fast Tree-Field Integrators are theoretically supported and experimentally verified.

**Strengths:**

The idea of IntegratorTree (IT) is original and new. The structure of IntegratorTree is carefully designed to reduce the computational complexity. The method is widely applicable for many practically used functions, not restricted to Hermitian or SDD. All the properties of Fast Tree-Field Integrators are theoretically supported and experimentally verified.

**Weaknesses:**

The paper seems to consider the machine learning community as potential readers, but the underlying concepts are fairly unknown to the machine learning community. The reviewer also does not do research in integrating tensor fields on graphs, and it was a bit difficult to follow the concepts and structures in Fast Tree-Field Integrators (FTFI). I think this presentation problem can be greatly improved by providing examples or figures. For example, Section 3.2 would have been more easily grasped if the authors provided pictorial examples as Figure 1.

========================================

This weakness about the presentation of FTFI is addressed by the rebuttal, so I am increasing my score from 6 to 7.

**Questions:**

In the experiments, the authors have only used minimum spanning tree (MST) as the tree approximation of a graph. On one side, this makes sense since the paper focuses on the fast computation of matrix-vector multiplication on a graph, not on choosing an appropriate tree. However, it seems to me that one reason of the relatively unsatisfactory accuracy (cosine similarity) of Fast Tree-Field Integrators (FTFI) is due to the distortion of the geometry of a graph by MST. As the authors have presented related work in embedding graphs to trees, I was wondering if we can get some improvements in accuracy when we use more advanced methods of embedding graphs to trees.

========================================

The question is addressed by the rebuttal.

**Limitations:**

The authos have addressed the limitations that Fast Tree-Field Integrators (FTFI) can work well provided that a graph is well approximated by a tree. I do not find any societal impact of this work.

---

> ### Author Rebuttal · Authors · 2024-08-06
>
> **General comment:**
>
> We would like to sincerely thank the Reviewer for the feedback.
>
> **Improved presentation with the pictorial example in Sec. 3.2 (as Fig. 1):**
>
> Thank you very much for the comment. Following Reviewer’s suggestion, we added a pictorial description of the divide-and-conquer method presented in Sec. 3.2. We have included it in the pdf attached to the rebuttal.
>
> **More advanced embeddings of graphs to trees for improved accuracy:**
>
> Thank you for an excellent comment. It is indeed the case that FTFI in principle can work with any tree, not necessarily only Minimum Spanning Tree (MST). However if the pre-processing time, necessary for tree-construction, is itself quadratic in the graph size, that defeats the purpose of using FTFI. For that reason, we focused on trees that can be efficiently constructed. MST is a perfect candidate since it can be build in log-linear time. Furthermore, near minimum spanning trees, often provide constant average distortion ([1]). We want to emphasize that for mesh-modeling experiments in Fig. 4, FTFI for several meshes performs similarly accuracy-wise to the brute-force algorithm (as we show in the third plot on one 3K-size mesh example). This might not be clearly seen in the second plot, since many dots overlap there. We will clarify it in the camera-ready version of the paper.
>
> We actually already applied more advanced tree embeddings of graphs for mesh-modeling, as shown in Fig. 4. We used in particular: (1) Bartal trees and ([2])  Fakcharoenphol trees (FRT, [3]), as well as non-tree embeddings leveraging small-separator factorization of mesh-graphs (SF), recently introduced in [4]. The Bartal tree approach provided some accuracy improvements over MST (the FTFI instantiation used in Fig. 4 applied MST), but at the price of 4x slower pre-processing time. The method applying Fakcharoenphol trees could be in practice applied only for very small meshes of sizes <=1K, due to the infeaibly large pre-processing time for larger graphs. The SF method was worse accuracy-wise than FTFI.
>
> We have also conducted additional experiments, applying FTFI to masked Transformers for video data. FTFI applied in a topological masking mechanism in the ViViT architecture ([5], factorized Transformer model for videos,  trained from scratch) leads to the **+0.8% absolute** improvement as compared on the Kinetics dataset ([6]). **To the best of our knowledge, this is the first application of the Topological Transformers for video data**. If instead of the MST, a full grid was applied for masking, the additional improvement was only negligible ( **+0.2%**). That shows that using other, more sophisticated graph-embeddings would also lead to only marginal additional improvements.
>
>
>
> [1] On notions of distortion and an almost minimum spanning tree with constant average distortion., Bartal et al., STOC 2016.
>
> [2] On approximating arbitrary metrics by tree metrics, Bartal, Y., STOC 1998
>
> [3] A tight bound on approximating arbitrary metrics by tree metrics, Fakcharoenphol et al., . J. Comput. Syst. Sci.
>
> [4] Efficient Graph Field Integrators Meet Point Clouds, Choromanski et al. ICML 2023.
>
> [5] ViViT: A Video Vision Transformer, Arnab et al., ICCV 2021.
>
> [6] The kinetics human action video dataset, Kay et al., 2017.

---

> > ### Author Response · Authors · 2024-08-09
> > **Addressing comments of Reviewer o1jE**
> >
> > Dear Reviewer o1jE,
> >
> > We would like to once more sincerely thank you for all the comments and very useful feedback. We think that we have addressed in depth all Reviewer's questions. Please let us know. If the Reviewer has any additional questions, we would be more than happy to answer them.
> >
> > Yours sincerely,
> >
> > The Authors

---

> > > ### Author Response · Authors · 2024-08-11
> > >
> > > Dear Reviewer o1jE,
> > >
> > > We would like to once more sincerely apologize for taking your time. As we mentioned before, we believe we have addressed all Reviewer’s comments. We do hope that the Reviewer can update the score correspondingly. If the Reviewer has any additional questions, please let us know and we will be happy to address them. Thank you very much !
> > >
> > > Yours sincerely,
> > >
> > > The Authors

---

> > > > ### Author Response · Authors · 2024-08-13
> > > >
> > > > Dear Reviewer o1jE,
> > > >
> > > > We would like to once more sincerely apologize to you for taking your time. We believe that we have addressed all Reviewer's questions, in particular: (1) provided additional figure explaining main algorithmic concepts (see: attached pdf) to improve presentation, (2) discussed results with more advanced trees (than minimum spanning trees), **which are actually already in the paper** (the Reviewer might have missed them so we clarified this point in the rebuttal). In addition, we have included in particular new experiments with Transformers on video input (as well as experiments with additional image datasets and speed tests). The experiments with this new data modality confirm all our findings that were already included in the paper.
> > > >
> > > > Since the end of the discussion period is approaching, we would like to sincerely ask the Reviewer to comment on the rebuttal and update the score accordingly. If the Reviewer has any additional questions, please let us know so that we can address them before the end of the discussion period.
> > > >
> > > > Once more, thank you very much for your reviews and feedback !
> > > >
> > > > Yours sincerely,
> > > >
> > > > The Authors

---

> > ### Comment · Reviewer_o1jE · 2024-08-14
> >
> > I would like to thank the authors for addressing weakness and questions I proposed. I think all of them are addressed, so I increased the score from 6 to 7.

---

### Official Review · Reviewer_Lag9 · 2024-07-22

**Soundness:** 3
**Presentation:** 3
**Contribution:** 2
**Rating:** 5
**Confidence:** 1

**Summary:**

The authors propose an algorithm for exact polylog-linear multiplication for general weighted trees and cordial functions f, which leads to a fast algorithm for distance-matrix tensor multiplication as used in transformers and graph kernels. The core of the algorithm is a binary tree structure called integration tree, which allows to perform the integration using an efficient divide and conquer scheme. The authors show that, for matrices defined by d-cordial functions, the integration can then be done by O(N log^{d+1} (N)). The authors show that their algorithm improves runtime against naive baselines in graph classification, vertex normal prediction, interpolation on meshes and topological vision transformers.

The work in general is far away from my research area. Thus, my confidence is low and I feel not qualified to judge the significance of  theoretical contributions in this work.

**Strengths:**

- The results suggests that the presented algorithm is clearly faster than brute force tree-field integration and that it does not negatively impact the quality of results on downstream tasks.
- The technical and theoretical contributions seem to be well thought and thorough. The algorithm makes sense to me on a high level.
- I reviewed this paper before and the presentation quality improved, making it a bit easier to understand how it embeds into related work and what its contributions are.

**Weaknesses:**

I agree with the authors that the tackled integration problem is relevant in many machine learning / deep learning models. It is unclear to me though how often the tree assumption is applicable. The practical experiments on topological vision transformers and graph classification, showing usefulness in practice. However, the experiments seem to be a bit inconclusive due to lacking trade-off analysis

What this paper lacks is an experiment that conclusively shows that the presented algorithm can be integrated into a state-of-the-art method (such as general masked transformers on multiple tasks) and that it has a positive impact on the quality/efficiency trade-off.

It would be also helpful if source code would be released in order to replicate/verify results.

**Questions:**

How does the quality vs. efficiency trade-off behave in the topological vision transformers experiment? It is only shown that the quality slightly improves but not if the method has an impact on transformer efficiency.

**Limitations:**

The authors briefly discuss limitations in the appendix.

---

> ### Author Rebuttal · Authors · 2024-08-06
>
> **General comment:**
>
> We would like to sincerely thank the Reviewer for the feedback.
> **We provide additional responses in the official comment titled: "Additional responses for Reviewer Lag9".**
>
> **A conclusive experiment showing integration into SOTA and positive impact on the quality/efficiency trade-off:**
>
> Thank you very much for the comment. Following Reviewer’s suggestion, we have decided to extend our experiments on efficient Transformers, to provide a **conclusive** evidence of the broad applicability of the method for Transformers. We target efficient-attention methods (an important field of the research on Transformers), since our paper focuses on **computational efficiency**. Thus comparing quality-wise with brute-force quadratic-attention Transformers is not relevant. **We show that FTFI leads to consistent accuracy improvements over SOTA efficient-attention Transformer models, with no speed loss or even speed gains**. For the comparison, we chose in particular low-rank attention Transformers due to their conceptual simplicity and the fact that they are broadly applied in various fields, including vision, speech, NLPs, Robotics, see: [1], [2], [3], [4], [5]. The interest in low-rank attention led also to designing hardware-efficient algorithms to optimize its on-device performance ([6]). We also compared against popular cosFormer architectures ([7]) and modified low-rank attention mechanism with gating (RF-Gate-Gaussian) from [4]. **We have also added extra experiments with video input data.**
>
> [1] Rethinking Attention with Performers, Choromanski et al., ICLR 2021.
>
> [2] Transformers are RNNs: Fast Autoregressive Transformers with Linear Attention, Katharopoulos et al., ICML 2020.
>
> [3] The Hedgehog & the Porcupine: Expressive Linear Attention with Softmax Mimicry, Zhang et al., ICLR 2024.
>
> [4] Random Feature Attention, Peng et al., ICLR 2021.
>
> [5] SARA-RT: Scaling up Robotics Transformers with Self-Adaptive Robust Attention, Leal et al., ICRA 2024, Bert Robotic Manipulation Paper Award.
>
> [6] Gated Linear Attention Transformers with Hardware-Efficient Training, Yang et al., ICML 2024.
>
> [7] cosFormer: Rethinking Softmax in Attention, Qin et al., ICLR 2022.
>
> [8] Learning a Fourier Transform for Linear Relative Positional Encodings in Transformers, Choromanski et al., AISTATS 2024.
>
> We present detailed comparison below.
>
> **1. ViTs - accuracy**
>
> **1.1 ImageNet:**
>
> We have already provided comparison with SOTA efficient-attention methods: low-rank attention Transformers in Sec 4.4, quality-wise.  On standard ImageNet benchmark, our best Transformer with FTFI provide **78.15%** accuracy, as compared to **76.37%** of the best low-rank -attention variant (obtained by testing three different linear variants). That gives **1.78%** accuracy improvement with only **3** extra trainable parameters per head (**36** extra trainable parameters per layer). To understand how significant **1.78%** accuracy improvement on ImageNet is, note that the technique of self-supervised pre-training objective (masked patched prediction, inspired by masked language modeling), widely considered as providing a **significant improvement** of regular ViTs and adopted in most ViT repositories, yet requiring a completely different training procedure, provides **2%** accuracy gain on ImageNet. For the rebuttal, we have also run the experiments with cosFormer. It achieved **76.3%** accuracy (consistent with what is reported in the literature, see [8]), lower than both: our method and the best tested low-rank attention variant. The RF-Gate-Gaussian achieved **76.35%** accuracy, which is is still lower than both: FTFI and the best tested low-rank attention variant.
>
>
> **1.2 Places365:**
>
> We have also conducted tests on another challenging dataset: Places365. In the paper, we report **1.71%** accuracy improvement over low-rank attention Transformer (**56.51%** accuracy vs **54.8%** accuracy). For the rebuttal, we also run the experiment with cosFormer which achieved **55.4%** accuracy (consistent with what is reported in the literature, see: [8]). This is still **0.93%** behind our method. The RF-Gate-Gaussian achieved accuracy **55.1%**, lower than this of cosFormer.
>
> **1.3. I-naturalist dataset:**
>
> I-naturalist is yet another challenging dataset, with **10K** classes, diverse image quality and significant class imbalance. Transformer with FTFI provides **1%** accuracy improvement over its regular low-rank attention counterpart and the cosFormer. Furthermore, FTFI achieved **0.8%** improvement over RF-Gate-Gaussian.  The convergence of the FTFI variant is **20-23%** faster than this of its regular low-rank attention counterpart, the cosFormer and RF-Gate-Gaussian.
>
> **2. ViTs - speed**
>
> In the rebuttal, we also add speed tests, to completely describe quality-efficiency trade-off. Our Transformer with FTFI is  as fast as regular low-rank attention Transformers and cosFormer, and 10% faster than RF-Gate-Gaussian. It provides **25%** speedup over regular brute-force quadratic-attention ViT.
>
> Those speedups are for a setting with a default number of patches used in the brute-force variants of the architectures. With more patches, speed gains are even more substantial. However then brute-force variants are too slow to train and thus we could not incorporate those results into our studies.
>
> **3. Masked Transformers for videos**
>
> We applied FTFI also for Transformers operating on videos. To the best of our knowledge, we are the first to use topological masking min that setting.
>
> FTFI applied in a topological masking mechanism in ViViT ([1], factorized Transformer model for videos,  trained from scratch) leads to the **+0.8% absolute** improvement as compared on the Kinetics dataset ([2]). **To the best of our knowledge, this is the first application of the Topological Transformers for videos**.
>
> [1] ViViT: A Video Vision Transformer, Arnab et al., ICCV 2021.
>
> [2] The kinetics human action video dataset, Kay et al., 2017.

---

> > ### Author Response · Authors · 2024-08-09
> > **Addressing comments of Reviewer Lag9**
> >
> > Dear Reviewer Lag9,
> >
> > We would like to once more sincerely thank you for all the comments and very useful feedback. We think that we have addressed in depth all Reviewer's questions. Please let us know. If the Reviewer has any additional questions, we would be more than happy to answer them.
> >
> > Yours sincerely,
> >
> > The Authors

---

> > > ### Author Response · Authors · 2024-08-11
> > >
> > > Dear Reviewer Lag9,
> > >
> > > We would like to once more sincerely apologize for taking your time. As we mentioned before, we believe we have addressed all Reviewer’s comments. We do hope that the Reviewer can update the score correspondingly. If the Reviewer has any additional questions, please let us know and we will be happy to address them. Thank you very much !
> > >
> > > Yours sincerely,
> > >
> > > The Authors

---

> > > > ### Author Response · Authors · 2024-08-13
> > > >
> > > > Dear Reviewer Lag9,
> > > >
> > > > We would like to once more sincerely apologize for taking your time. **We believe we have addressed all Reviewer's comments**. In particular, we have conducted a comprehensive set of additional experiments, **showing conclusively** that Transformers with FTFI are SOTA efficient Transformers methods for image data. We also included new experiments with other modality. To be more specific, we have run additional experiments with Transformers on new image dataset (I-naturalist) as well as on a new modality, namely: videos. In those experiments we also tested many other efficient Transformer-architectures. These additional experiments confirm all our previous findings. In particular, FTFI applied in a topological masking mechanism in the ViViT architecture ([1], factorized Transformer model for videos, trained from scratch) leads to the +0.8% absolute improvement as compared on the Kinetics dataset ([2]). To the best of our knowledge, this is the first application of the Topological Transformers for video data.
> > > >
> > > > [1] ViViT: A Video Vision Transformer, Arnab et al., ICCV 2021.
> > > >
> > > > [2] The kinetics human action video dataset, Kay et al., 2017.
> > > >
> > > > For image modality, we have conducted a comprehensive comparison, not only with one linear-attention Transformer-model, but with **three different classes** of efficient Transformer-architectures, namely: (1) linear-attention methods based on low-rank decomposition (we tested several different instantiations with different attention-kernel functions) ([3]), (2) cosFormers ([4]) and (3) RF-Gate-Gaussian models ([5]). The FTFI-Transformer outperforms all of them quality-wise, while being at least as fast as those methods.
> > > >
> > > > [3] Rethinking Attention with Performers, Choromanski et al., ICLR 2021.
> > > >
> > > > [4] cosFormer: Rethinking Softmax in Attention, Qin et al., ICLR 2022.
> > > >
> > > > [5] Gated Linear Attention Transformers with Hardware-Efficient Training, Yang et al., ICML 2024.
> > > >
> > > >
> > > > We thus would like to sincerely ask the Reviewer to comment on the rebuttal and update the score accordingly. If the Reviewer has any additional questions please let us know and we will be happy to answer them. **In particular, if there are any other specific experiments regarding Transformers experiments on image data the Reviewer would like to see, please let us know which specific experiments the Reviewer would like us to conduct and we will be more than happy to do it**. The discussion period is almost over thus we want to make sure that all additional comments the Reviewer will have (if any) can be promptly addressed by us.
> > > >
> > > > Once more, thank you very much for your comments and feedback !
> > > >
> > > > Yours sincerely,
> > > >
> > > > The Authors

---

### Official Review · Reviewer_SX39 · 2024-07-22

**Soundness:** 4
**Presentation:** 4
**Contribution:** 3
**Rating:** 7
**Confidence:** 4

**Summary:**

This paper proposes faster matrix multiplication algorithms for a class of structured matrices, namely, f-distance matrices, where the $(i, j)^{th}$ entry of the matrix is $f(\text{dist}(i, j))$, for nodes $i, j$ in a graph. The authors propose an algorithm, referred to as Fast Tree-Field Integrators (FTFI), for the case of tree graphs (and in the experiments, for general graphs, the graph is approximated by its spanning tree).

The FTFI algorithm works as follows: first, an "integrator tree" is constructed (which is created only once for the f-distance matrix, and reused whenever this matrix is multiplied by another matrix). The integrator tree is a binary tree where each node is a subtree of the original tree. To obtain the two children of a node, they pick some vertex in the subtree (of the original tree) that this node corresponds to - this node will be called the "pivot point", and the subtree will be split into a "left" and "right" subtree at this pivot point.

This integrator tree is used to recursively compute the product of the $f$-distance matrix (denoted $M^G_f$) with another arbitrary matrix/tensor $X$. To compute the row of $M^G_f X$ corresponding to the vertex $v$, suppose $v$ is in the left subtree of the current node of the integrator tree. Then, we have to consider the contribution of rows of $X$ from vertices $j$ that are also in the left subtree (which is dealt with recursively) and vertices $j$ that are in the right subtree. In the latter case, the entry $f(dist(v, j))$ of the $f$-distance matrix can be written as $f(dist(v, p) + dist(p, j))$ where $p$ is the pivot point, and this property is leveraged to achieve a running time faster than naive matrix multiplication in several cases, such as when $f$ is a rational function or an exponential function.

The authors compare FTFI to BTFI (brute-force matrix multiplication in the tree setting), and other algorithms including BGFI (brute-force matrix multiplication without approximating the graph by a tree). The tasks studied include interpolation on meshes and graph classification. For interpolation on meshes, FTFI is the fastest in terms of preprocessing. It performs similarly to the SF algorithm (a state of the art algorithm for this problem), and performs worse than BGFI, which is a more brute force, but more accurate, algorithm. In the graph classification setting, FTFI gets similar accuracy as BGFI while being significantly faster.

The authors finally do experiments with FTFI on topological vision transformers. In a large scale setting, FTFI obtains a 7% improvement on ImageNet compared to the standard ViT performer (which obtains around 70% accuracy).

**Strengths:**

- The matrices studied in this work are far more general than those studied by previous works (structured matrices studied in previous works are exponentials of adjacency matrices, symmetric diagonally dominant matrices, power series of random walk kernels, boolean matrices).
- The algorithm is interesting technically - decomposing the entries of the f-distance matrix to avoid the running time of naive matrix multiplication is a good idea.
- The results are strong, particularly on graph classification.

**Weaknesses:**

- There are some questions I have about the experimental results, mentioned below in the questions section. If these questions are addressed, I would be willing to raise the score.

**Questions:**

- Why does BTFI require more time for processing than BGFI in Figure 4? Intuitively, they would require similar effort, and BGFI should perhaps be slower (with computing the shortest paths potentially having a longer running time for more general graphs than for trees).
- Also, given that BGFI does significantly better than BTFI/FTFI in terms of cosine similarity, at the cost of a roughly 5x slowdown compared to FTFI according to Figure 4, is it better to use BGFI compared to FTFI? What are the considerations here?
- Do the plots on the right-hand side of Figure 4 contradict the plots on the left-hand side of Figure 4? The second plot from the left in Figure 4 suggests that BGFI generally achieves greater cosine similarity than FTFI for various numbers of points (though it is slower) while the third plot from the left shows that FTFI is both faster and achieves higher cosine similarity than BGFI. How do you interpret the difference?
- In Figure 5, on PTC-MR, why is BGFI slightly faster than FTFI? Is this a specific distribution of graphs?
- For the experiments in Figure 6, the goal is to show that tree-based estimators can emulate integration on arbitrary graphs, using a rational function with quadratic numerator and denominator. I am not convinced that this experiment shows that tree-based metrics can emulate general graph metrics, since the MSE loss seems to plateau around 1. This seems a bit large given that the distribution of weights is taken from (0, 1). What is the distribution of graphs?

========================================

These questions are addressed by the rebuttal, and I am increasing my score from 6 to 7.

**Limitations:**

Yes.

---

> ### Author Rebuttal · Authors · 2024-08-06
>
> **General comment:**
>
> We would like to sincerely thank the Reviewer for the feedback.
>
> **Processing time of BTFI vs BGFI in Fig. 4:**
>
> Thank you very much for an excellent comment. We have realized that for BTFI we unnecessary applied Kruskal algorithm two times to compute the minimum spanning tree. After the correction, BTFI pre-processing time is shorter than before, in particular a little shorter than this of the BGFI algorithm. FTFI is still the fastest and SF is still second fastest. Thus our claims remain the same. In the pdf attached to the rebuttal, we put in particular corrected first plot of Fig. 4. We will also put corrected Fig. 4 in the camera-ready version of the paper. Once more, thank you for catching this !
>
>
> **BGFI vs FTFI:**
>
> The FTFI algorithm is designed to scale-up to massive graphs, where **5x** speedup differentiates between a feasible and non-feasible processing time. FTFI is the fastest method among all six tested algorithms (achieving 2.5x+ speedup over the second fastest method in Fig. 4), obtaining similar or better accuracy than other tested fast algorithms. Thus we recommend to apply FTFI for massive graphs, where the processing time of BGFI is not acceptable. Furthermore, for several meshes FTFI matches quality-wise BGFI, yet it might be hard to see that on the second plot, since the corresponding dots overlap (this is clearly illustrated in particular on the third plot; see also: our discussion on second vs third plot in Fig. 4).
>
> We also would like to emphasize that we showed in Sec. 4.4 that FTFI is an option for Topological Transformers to provide accuracy-gains over regular ViTs. The ability to process massive graphs translates in this setting to processing high-resolution images or images partitioned into small patches (e.g. for pixel-to-pixel attention).
>
>
> **Second plot vs the third plot in Fig. 4:**
>
> Thank you very much for the comment. Even though computational advantages of FTFI for mesh-graphs from Thingi10K dataset considered in the first part of Sec. 4.2 in general imply some accuracy loss as compared to the brute-force baseline, the gap depends a lot on the graph structure (this observation is aligned with the voluminous literature on the quality of tree-metric based approximation and a subject on its own). To show it, we created third and fourth plot in Fig.4 where we sampled two meshes of sizes 3K (the actually mesh size is ~2.7K) and 5K respectively (as described in the caption of Fig. 4). In the third plot, we demonstrate that the gap might not even exist. These were not “cherry-picked examples”. To see that note that even though in the third plot FTFI matches quality-wise BGFI, in the fourth plot we show that BGFI is the best accuracy-wise (with FTFI being second-best and the fastest). To sum it up, the role of the last two plots in Fig. 4 was to focus on a particular mesh (per plot) to make presentation more clean. Note that in the second plot lots of dots overlap due to different methods performing similarly accuracy-wise and thus some well-performing FTFI variants as the 3K-one can be easily missed. We will clarify it in the final version of the paper and improve second plot to make it more clear.
>
> To be very specific, the sampled 3K- and 5K-size meshes from the Thingi10K dataset were of ids: 1514901 and 39463 respectively.
>
>
> **BGFI vs FTFI speed-wise in Fig. 5; PTC-MR:**
>
> The Reviewer is correct, this is due to a particular structure of those graphs and the fact that they are very small (average number of nodes is only **14.29**, as shown in Table 2 in the Appendix).
>
> **Experiments in Fig. 6:**
>
> We would like to emphasize that even though graphs’ weights in these experiments are taken from the interval (0, 1), the entries of the matrices under consideration are much larger since they encode distances of the paths. Thus in practice they can be of order a few hundred, since considered graphs have **800** nodes. SOTA non-learnable, purely algorithmic approaches based on low-distortion trees, such as Bartal trees ([1]) or Fakcharoenphol trees ([2]), providing log(N)lo log(N) and log(N) distortion ratio respectively, would lead to errors 9.0+. The goal of this experiment was to show that by learning functions of distances, this loss can be dramatically decreased, even if the underlying spanning tree is a minimum spanning tree (that can be quickly computed). Furthermore,  we showed that training can be done in sub-quadratic time by sampling a compact set of source-target points and computing shortest-path distances between them. Note that we used a relatively simplistic training setting with only **six** trainable parameters since function f was parameterized as a ratio of two quadratic functions.
>
> As we explained in the caption of Fig.6, the graphs are obtained from a path on 800 nodes, by adding 600 random edges and assigning independent weights to them from the interval (0, 1). Thus considered graphs are sparse.
>
> [1] On approximating arbitrary metrics by tree metrics, Bartal, Y., STOC 1998
>
> [2] A tight bound on approximating arbitrary metrics by tree metrics, Fakcharoenphol et al., . J. Comput. Syst. Sci.

---

> > ### Comment · Reviewer_SX39 · 2024-08-13
> >
> > Thank you for your reply, which clarifies my questions - I will increase my score. In the final version, it would indeed be useful to have a version of the second plot in Figure 4 where the points do not overlap. The explanation of Figure 6 is helpful, since it gives a sense of what error can be typically achieved using tree-based metrics.

---

> ### Author Response · Authors · 2024-08-09
> **Addressing comments of Reviewer SX39**
>
> Dear Reviewer SX39,
>
> We would like to once more sincerely thank you for all the comments and very useful feedback. We think that we have addressed in depth all Reviewer's questions. Please let us know. If the Reviewer has any additional questions, we would be more than happy to answer them.
>
> Yours sincerely,
>
> The Authors

---

> > ### Author Response · Authors · 2024-08-11
> >
> > Dear Reviewer SX39,
> >
> > We would like to once more sincerely apologize for taking your time. As we mentioned before, we believe we have addressed all Reviewer’s comments. We do hope that the Reviewer can update the score correspondingly. If the Reviewer has any additional questions, please let us know and we will be happy to address them. Thank you very much !
> >
> > Yours sincerely,
> >
> > The Authors

---

### Author Response · Authors · 2024-08-06
**General comment**

We would like to sincerely thank all the Reviewers for their very valuable feedback and comments. We addressed all of them in the rebuttals below and additional official responses. We also want to emphasize that we provide the pointer to the anonymous repository with the code:

https://anonymous.4open.science/r/FastTreeFieldIntegrator/README.md.

That repository contains in particular: **detailed implementation** of the FTFI algorithm (key contribution of the paper) with several unit tests, as well as the code for graph classification and vertex-normal prediction tasks (we will release any remaining code upon the acceptance of the paper).

Finally, we have also run additional experiments with Transformers on new image dataset (I-naturalist) as well as on a  **new modality**, namely: videos. In those experiments we also test many other efficient Transformer-architectures. These additional experiments **confirm all our previous findings**. In particular, FTFI applied in a topological masking mechanism in the ViViT architecture ([1], factorized Transformer model for videos,  trained from scratch) leads to the **+0.8% absolute** improvement as compared on the Kinetics dataset ([2]). **To the best of our knowledge, this is the first application of the Topological Transformers for video data**.

[1] ViViT: A Video Vision Transformer, Arnab et al., ICCV 2021.

[2] The kinetics human action video dataset, Kay et al., 2017.

We also conducted additional speed tests for Transformers to describe speed-quality trade-offs, showing that our methods are as fast as previous efficient-attention Transformers, while providing better quality.

---

> ### Author Response · Authors · 2024-08-06
> **Additional responses for Reviewer H9MT**
>
> **"How do you turn the general graph into a tree ?". Information lost and theoretical analysis**:
>
> Thank you very much for the comment. In practice, we construct minimum spanning trees (MSTs) for those graphs. In principle, trees providing low distortion of the general graph metric can be used (the best algorithmic constructions provide log(N)-distortion ratio, where N is the number of nodes of the graph). Low distortion trees is a subject of an extensive research, as we show in in Sec. 2 and Appendix B. However with learnable functions f applied for the shortest-path lengths in the trees, the approximation guarantees can be substantially improved, as we demonstrate in Sec. 4.3. We provide additional theoretical discussion (also regarding near minimum spanning trees) in other responses to Reviewer H9MT, in particular below.
>
> **Theoretical guarantees for almost-tree like graphs:**
>
> This is a second part of the response to Reviewer H9MT from the paragraph: **FTFI to approximate almost-tree like graphs and general graphs: theoretical analysis** in the rebuttal for Reviewer H9MT below.
>
> Following Reviewer’s comment, we would like to give examples of the almost-tree like graphs, where our method can be applied to produce integration of arbitrary precision in polylog-linear time, as long as all edge-weights are taken from the bounded interval. We also provide detailed sketch of the proof below and will add the full proof in the camera-ready version of the paper.
>
> The class of graphs we consider are called bounded connected treewidth graphs, e.g. the graphs that admit a treewidth decomposition with all bags being connected graphs of bounded size. We refer the Reviewer to [1] for an introduction to the theory of treewidth.
> To see that these are natural candidates for almost tree-like graphs, note that in the treewidth decomposition of trees every bag is a graph corresponding to an edge (thus connected and of two nodes). It turns out that a treewidth decomposition of graphs with bounded connected treewidth, where all bags are connected and bounded can be found in linear time.
> Furthermore, in such graphs a balanced separators of constant size can be found in linear time (it can be found among bags of the treewidth decomposition). Now note that the key trick of the FTFI algorithm is a divide-and-conquer strategy, where a “pivot node” is found that splits the tree into two substantial parts. For bounded connected treewidth graphs, this node is simply replaced by the balanced separator. In the regular FTFI algorithm, vertices are then partitioned based on the distance from the pivot node. For bounded connected treewidth graphs, we instead apply two levels of partitioning. For every vertex, we first register its shortest path distance d to the balanced separator (defined as the distance to the closest vertex of the separator) and then we compute its so-called signature vector (d1,...,ds), where d+di is the shortest distance to the ith vertex of the separator and s is the number of all vertices of the separator. Note that since the separator is of constant size and connected and furthermore the weights are taken from the bounded set, (d1,...,ds) is taken from the bounded s-dimensional cube C. We then construct an epsilon-net N to cover that cube and assign for each vertex of the graph the node of the epsilon-net that is closest to its signature vector. Note that the number of nodes of the epsilon-net is constant. To effectively conduct the divide-and-conquer strategy in the setting with three sets of vertices: A, S and B, where S is s separator that separates A and B, we proceed as follows. We choose a pair of nodes (n1, n2) in the epsilon-net N. We then consider vertices from A for which n1 was assigned and vertices from B for which n2 was assigned (first level of partitioning). We then partition those vertices based on the distance from the separator S (second level of partitioning). Then the corresponding cross-terms can be computed in the completely analogous way as from the analysis in Sec. 3.2. We repeat this procedure for all pairs of nodes from the epsilon-net N. That completes the sketch of the proof since the size of the epsilon-net N is constant, and thus the number of all the pairs of nodes of the epsilon-net N is also constant.
>
> [1] Parameterized Algorithms, Pilipczuk et al., Springer 2015.

---

> ### Author Response · Authors · 2024-08-06
> **Additional responses for Reviewer Lag9**
>
> **“It is unclear to me though how often the tree assumption is applicable”:**
>
> Thank you for the comment. The quality of the general graph metric approximation with tree-metrics is a research area on its own, with voluminous corresponding literature. We provided a summary in Sec. 2 as well as in Appendix B. Notably, tree-metrics is a well-established tool, used in several applications, e.g. in distributed & online algorithms and biology (see for example: [1,2,3]). Theoretical analysis of tree-metrics is beyond the scope of this paper. In the paper we show that those tools can be effectively applied in machine learning. Rather than cherry-picking particular applications, where tree-metrics might be relevant, we provide extensive experimental evaluations for: graph classification, mesh modeling and Transformer training, where those methods were (to the best of our knowledge) never used before, showing the generality of the approach.
>
> Furthermore, in Sec. 4.3 we provide a clear evidence that machine learning provides yet another path for effectively applying tree-metrics. We show that by applying relatively simple learnable nonlinear functions f (ratios of quadratic functions; only **six** trainable parameters in total) on the shortest path distances in the minimum spanning tree, one can substantially improve the quality of the approximation of the general graph metrics. That results in **9x+** more accurate approximation than this given by classic SOTA algorithmic and non-learnable approaches, such as Bartal trees ([4]) or Fakcharoenphol trees ([5]), providing log(N)lo log(N) and log(N) distortion ratio respectively. Importantly, as we show in this paper, integration with those nonlinear functions f can be done efficiently, in polylog-linear time. Thus we showed that machine learning methods can be effectively applied to improve the accuracy of the approximation with tree-metrics. To the best of our knowledge, this is the first such result.
>
> [1] Efficient distributed approximation algorithms via probabilistic tree embeddings, Khan et al.,
>      PODC 2008.
>
> [2] K-server via multiscale entropic regularization, Bubeck et al., STOC 2018
>
> [3] Distorted metrics on trees and phylogenetic forests, Mossel et al., IEEE ACM Trans. Comput.
>      Biol. Bioinform, 2007.
>
> [4] On approximating arbitrary metrics by tree metrics, Bartal, Y., STOC 1998
>
> [5] A tight bound on approximating arbitrary metrics by tree metrics, Fakcharoenphol et al., . J. Comput. Syst. Sci.
>
> **Source code:**
>
> Thank you very much for the comment. Following Reviewer’s suggestion, we provide the pointer to the anonymous repository with the code:
>
> https://anonymous.4open.science/r/FastTreeFieldIntegrator/README.md.
>
> That repository contains in particular: **detailed implementation** of the FTFI algorithm (key contribution of the paper) with several unit tests, as well as the code for graph classification and vertex-normal prediction tasks (we will release any remaining code upon the acceptance of the paper).

---

### Author Rebuttal · Authors · 2024-08-06

**Additional pdf with updated plot for Fig. 4 and an additional visualization for the paper**

We would like to sincerely thank all the Reviewers for the very valuable comments and feedback. We summarize our rebuttals in the official comment below and then provide responses to the individual questions of the reviewers in the rebuttals and additional official comments. Here we also attach the pdf with the updated plot for Fig. 4 and an additional visualization for the paper.

---

### Decision · Program_Chairs · 2024-09-25

**Decision:**

Accept (poster)

**Comment:**

All reviewers argue for acceptance, albeit with partial low scores. The improvements the paper has made are noted as well as the interesting theoretical contributions. The paper meets the bar for acceptance.